# Multi-task Self-supervised Graph Neural Networks Enable Stronger Task Generalization

**Mingxuan Ju**[1]**, Tong Zhao**[2]**, Qianlong Wen**[1]**, Wenhao Yu**[1]**,**
**Neil Shah**[2]**, Yanfang Ye**[1§]**, Chuxu Zhang**[3§]
[1]University of Notre Dame, [2]Snap Inc., [3]Brandeis University
[1]{mju2,yye7}@nd.edu; [2]{tzhao,nshah}@snap.com;
[3]chuxuzhang@brandeis.edu

## Abstract

Self-supervised learning (SSL) for graph neural networks (GNNs) has attracted increasing attention from the graph machine learning community in recent years, owing to its capability to learn performant node embeddings without costly label information. One weakness of conventional SSL frameworks for GNNs is that they learn through a single philosophy, such as mutual information maximization or generative reconstruction. When applied to various downstream tasks, these frameworks rarely perform equally well for every task, because one philosophy may not span the extensive knowledge required for all tasks. To enhance the task generalization across tasks, as an important first step forward in exploring *fundamental graph models*, we introduce PARETOGNN, a multi-task SSL framework for node representation learning over graphs. Specifically, PARETOGNN is self-supervised by manifold pretext tasks observing multiple philosophies. To reconcile different philosophies, we explore a multiple-gradient descent algorithm, such that PARETOGNN actively learns from every pretext task while minimizing potential conflicts. We conduct comprehensive experiments over four downstream tasks (i.e., node classification, node clustering, link prediction, and partition prediction), and our proposal achieves the best overall performance across tasks on 11 widely adopted benchmark datasets. Besides, we observe that learning from multiple philosophies enhances not only the task generalization but also the single task performances, demonstrating that PARETOGNN achieves better task generalization via the disjoint yet complementary knowledge learned from different philosophies. Our code is publicly available at https://github.com/jumxglhf/ParetoGNN.

## 1 Introduction

Graph-structured data is ubiquitous in the real world (McAuley et al., 2015; Hu et al., 2020). To model the rich underlying knowledge for graphs, graph neural networks (GNNs) have been proposed and achieved outstanding performance on various tasks, such as node classification (Kipf & Welling, 2016a; Hamilton et al., 2017), link prediction (Zhang & Chen, 2018; Zhao et al., 2022b), node clustering (Bianchi et al., 2020; You et al., 2020b), etc. These tasks form the archetypes of many real-world practical applications, such as recommendation systems (Ying et al., 2018; Fan et al., 2019), predictive user behavior models (Pal et al., 2020; Zhao et al., 2021a; Zhang et al., 2021a).

Existing works for graphs serve well to make progress on narrow experts and guarantee their effectiveness on mostly one task or two. However, given a graph learning framework, its promising performance on one task may not (and usually does not) translate to competitive results on other tasks. Consistent task generalization across various tasks and datasets is a significant and well-studied research topic in other domains (Wang et al., 2018; Yu et al., 2020). Results from the Natural Language Processing (Radford et al., 2019; Sanh et al., 2021) and Computer Vision (Doersch & Zisserman, 2017; Ni et al., 2021) have shown that models enhanced by self-supervised learning (SSL) over multiple pretext tasks observing diverse philosophies can achieve strong task generalization and learn intrinsic patterns that are transferable to multiple downstream tasks. Intuitively, SSL over multiple pretext tasks greatly reduces the risk of overfitting (Baxter, 1997; Ruder, 2017), because

---

§ Corresponding Author.

learning intrinsic patterns that well-address difficult pretext tasks is non-trivial for only one set of parameters. Moreover, gradients from multiple objectives regularize the learning model against extracting task-irrelevant information (Ren & Lee, 2018; Ravanelli et al., 2020), so that the model can learn multiple views of one training sample.

Nonetheless, current state-of-the-art graph SSL frameworks are mostly introduced according to only one pretext task with a single philosophy, such as mutual information maximization (Velickovic et al., 2019; Zhu et al., 2020; Thakoor et al., 2022), whitening decorrelation (Zhang et al., 2021b), and generative reconstruction (Hou et al., 2022). Though these methods achieve promising results in some circumstances, they usually do not retain competitive performance for all downstream tasks across different datasets. For example, DGI (Velickovic et al., 2019), grounded on mutual information maximization, excels at the partition prediction task but underperforms on node classification and link prediction tasks. Besides, GRAPHMAE (Hou et al., 2022), based on feature reconstruction, achieves strong performance for datasets with powerful node features (e.g., graph topology can be inferred simply by node features (Zhang et al., 2021d)), but suffers when node features are less informative, which is empirically demonstrated in this work. To bridge this research gap, we ask:

> ***How to combine multiple philosophies to enhance task generalization for SSL-based GNNs?***

A very recent work, AUTOSSL (Jin et al., 2022), explores this research direction by reconciling different pretext tasks by learning different weights in a joint loss function so that the node-level pseudo-homophily is promoted. This approach has two major drawbacks: (i) Not all downstream tasks benefit from the homophily assumption. In experimental results shown by Jin et al. (2022), we observe key pretext tasks (e.g., DGI based on mutual information maximization) being assigned zero weight. However, our empirical study shows that the philosophies behind these neglected pretext tasks are essential for the success of some downstream tasks, and this phenomenon prevents GNNs from achieving better task generalization. (ii) In reality, many graphs do not follow the homophily assumption (Pei et al., 2019; Ma et al., 2021). Arguably, applying such an inductive bias to heterophilous graphs is contradictory and might yield sub-optimal performance.

In this work, we adopt a different perspective: we remove the reliance on the graph or task alignment with homophily assumptions while self-supervising GNNs with multiple pretext tasks. During the self-supervised training of our proposed method, given a single graph encoder, all pretext tasks are simultaneously optimized and dynamically coordinated. We reconcile pretext tasks by dynamically assigning weights that promote the Pareto optimality (Désidéri, 2012), such that the graph encoder actively learns knowledge from every pretext task while minimizing conflicts. We call our method PARETOGNN. Overall, our contributions are summarized as follows:

- We investigate the problem of task generalization on graphs in a more rigorous setting, where a good SSL-based GNN should perform well not only over different datasets but also at multiple distinct downstream tasks simultaneously. We evaluate state-of-the-art graph SSL frameworks in this setting and unveil their sub-optimal task generalization.

- To enhance the task generalization across tasks, as an important first step forward in exploring fundamental graph models, we first design five simple and scalable pretext tasks according to philosophies proven to be effective in the SSL literature and propose PARETOGNN, a multi-task SSL framework for GNNs. PARETOGNN is simultaneously self-supervised by these pretext tasks, which are dynamically reconciled to promote the Pareto optimality, such that the graph encoder actively learns knowledge from every pretext task while minimizing potential conflicts.

- We evaluate PARETOGNN along with 7 state-of-the-art SSL-based GNNs on 11 acknowledged benchmarks over 4 downstream tasks (i.e., node classification, node clustering, link prediction, and partition prediction). Our experiments show that PARETOGNN improves the overall performance by up to +5.3% over the state-of-the-art SSL-based GNNs. Besides, we observe that PARETOGNN achieves SOTA single-task performance, proving that PARETOGNN achieves better task generalization via the disjoint yet complementary knowledge learned from different philosophies.

## 2 MULTI-TASK SELF-SUPERVISED LEARNING VIA PARETOGNN

In this section, we illustrate our proposed multi-task self-supervised learning framework for GNNs, namely PARETOGNN. As Figure 1 illustrates, PARETOGNN is trained with different SSL tasks

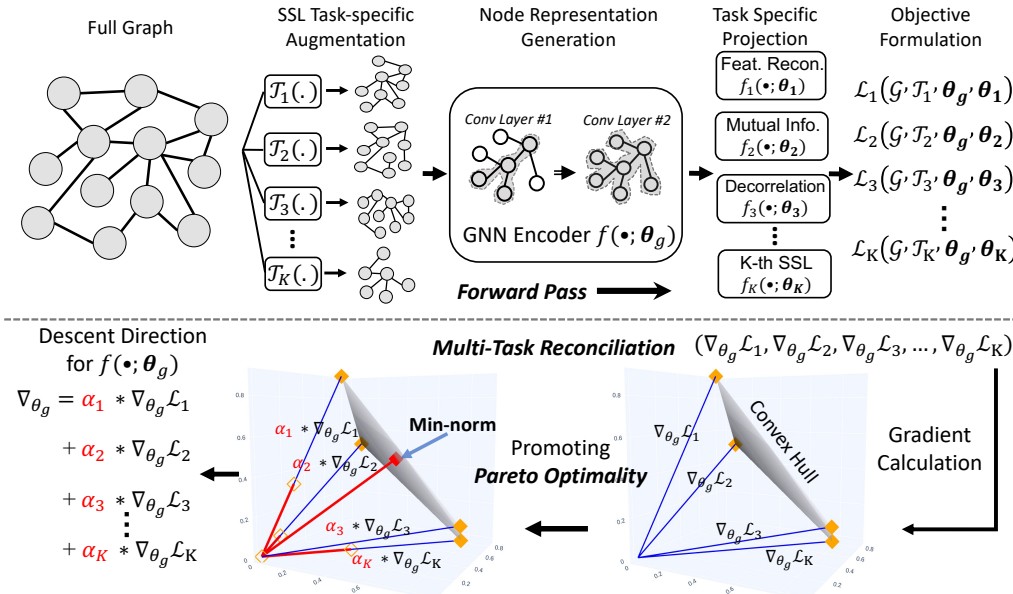

Figure 1: PARETOGNN is simultaneously self-supervised by $K$ SSL tasks. All SSL tasks share the same GNN encoder and have their own projection heads (i.e., $f_k(\cdot; \boldsymbol{\theta}_k)$) and augmentations (i.e., $\mathcal{T}_k(\cdot)$) such as node or edge dropping, feature masking, and graph sampling. To dynamically reconcile all tasks, we assign weights to tasks such that the combined descent direction has a minimum norm in the convex hull, which promotes Pareto optimality and further enhances task generalization.

simultaneously to enhance the task generalization. Specifically, given a graph $\mathcal{G}$ with $N$ nodes and their corresponding $D$-dimensional input features $\mathbf{X} \in \mathbb{R}^{N \times D}$, PARETOGNN learns a GNN encoder $f_g(\cdot; \boldsymbol{\theta}_g) : \mathcal{G} \rightarrow \mathbb{R}^{N \times d}$ parameterized by $\boldsymbol{\theta}_g$, that maps every node in $\mathcal{G}$ to a $d$-dimensional vector (s.t. $d \ll N$). The resulting node representations should retain competitive performance across various downstream tasks without any update on $\boldsymbol{\theta}_g$. With $K$ self-supervised tasks, we consider the loss function for $k$-th SSL task as $\mathcal{L}_k(\mathcal{G}; \mathcal{T}_k, \boldsymbol{\theta}_g, \boldsymbol{\theta}_k) : \mathcal{G} \rightarrow \mathbb{R}^+$, where $\mathcal{T}_k$ refers to the graph augmentation function required for $k$-th task, and $\boldsymbol{\theta}_k$ refers to task-specific parameters for $k$-th task (e.g., MLP projection head and/or GNN decoder). In PARETOGNN, all SSL tasks are dynamically reconciled by promoting Pareto optimality, where the norm of gradients w.r.t. the parameters of our GNN encoder $\boldsymbol{\theta}_g$ is minimized in the convex hull. Such gradients guarantee a descent direction to the Pareto optimality, which enhances task generalization while minimizing potential conflicts.

## 2.1 MULTI-TASK GRAPH SELF-SUPERVISED LEARNING

PARETOGNN is a general framework for multi-task self-supervised learning over graphs. We regard the full graph $\mathcal{G}$ as the data source; and for each task, PARETOGNN is self-supervised by sub-graphs sampled from $\mathcal{G}$, followed by task-specific augmentations (i.e., $\mathcal{T}_k(\cdot)$). The rationale behind the exploration of sub-graphs is two-fold. Firstly, the process of graph sampling is naturally a type of augmentation (Zeng et al., 2019) by enlarging the diversity of the training data. Secondly, modeling over sub-graphs is more memory efficient, which is significant especially under the multi-task scenario. In this work, we design five simple pretext tasks spanning three high-level philosophies, including generative reconstruction, whitening decorrelation, and mutual information maximization. However we note that PARETOGNN is not limited to the current learning objectives and the incorporation of other philosophies is a straightforward extension. Three high-level philosophies and their corresponding five pretext tasks are illustrated as follows:

- **Generative reconstruction**. Recent studies (Zhang et al., 2021d; Hou et al., 2022) have demonstrated that node features contain rich information, which highly correlates to the graph topology. To encode node features into the representations derived by PARETOGNN, we mask the features of a random batch of nodes, forward the masked graph through the GNN encoder, and reconstruct the masked node features given the node representations of their local sub-graphs (Hou et al., 2022). Furthermore, we conduct the similar reconstruction process for links between the connected nodes to retain the pair-wise topological knowledge (Zhang & Chen, 2018). Feature and topology reconstruction are denoted as `FeatRec` and `TopoRec` respectively.

- **Whitening decorrelation**. SSL based on whitening decorrelation has gained tremendous attention, owing its capability of learning representative embeddings without prohibitively expensive negative pairs or offline encoders (Ermolov et al., 2021; Zbontar et al., 2021). We adapt the same philosophy to graph SSL by independently augmenting the same sub-graph into two views, and then minimize the distance between the same nodes in the two views while enforcing the feature-wise covariance of all nodes equal to the identity matrix. We denote this pretext as `RepDecor`.

- **Mutual Information Maximization**. Maximizing the mutual information between two corrupted views of the same target has been proved to learn the intrinsic patterns, as demonstrated by deep infomax-based methods (Bachman et al., 2019; Velickovic et al., 2019) and contrastive learning methods (Hassani & Khasahmadi, 2020; Zhu et al., 2020). We maximize the local-global mutual information by minimizing the distance between the graph-level representation of the intact sub-graph and its node representations, while maximizing the distance between the former and the corrupted node representations. Besides, we also maximize the local sub-graph mutual information by maximizing similarity of representations of two views of the sub-graph entailed by the same anchor nodes, while minimizing the similarities of the representations of the sub-graphs entailed by different anchor nodes. The pretext tasks based on node-graph mutual information and node-subgraph mutual information are denoted as `MI-NG` and `MI-NSG`, respectively.

Technical details and objective formulation of these five tasks are provided in Appendix B. As described above, pretext tasks under different philosophies capture distinct dimensions of the same graph. Empirically, we observe that simply combining all pretext SSL tasks with weighted summation can sometimes already lead to better task generalization over various downstream tasks and datasets. Though promising, according to our empirical studies, such a multi-task self-supervised GNN falls short on some downstream tasks, if compared with the best-performing experts on these tasks. This phenomenon indicates that with the weighted summation there exist potential conflicts between different SSL tasks, which is also empirically shown by previous works from other domains such as Computer Vision (Sener & Koltun, 2018; Chen et al., 2018).

## 2.2 Multi-task Graph SSL promoting Pareto Optimality

To mitigate the aforementioned problem and simultaneously optimize multiple SSL tasks, we can derive the following empirical loss minimization formulation as:

$$\min_{\substack{\boldsymbol{\theta}_g, \\ \boldsymbol{\theta}_1, \ldots, \boldsymbol{\theta}_K}} \sum_{k=1}^{K} \alpha_k \cdot \mathcal{L}_k(\mathcal{G}; \mathcal{T}_k, \boldsymbol{\theta}_g, \boldsymbol{\theta}_k), \tag{1}$$

where $\alpha_k$ is the task weight for $k$-th SSL task computed according to pre-defined heuristics. For instance, AUTOSSL (Jin et al., 2022) derives task weights that promote pseudo-homophily. Though such a formulation is intuitively reasonable for some graphs and tasks, heterophilous graphs, which are not negligible in the real world, directly contradict the homophily assumption. Moreover, not all downstream tasks benefit from such homophily assumption, which we later validate in the experiments. Hence, it is non-trivial to come up with a unified heuristic that suits all graphs and downstream tasks. In addition, weighted summation of multiple SSL objectives might cause undesirable behaviors, such as performance instabilities entailed by conflicting objectives or different gradient scales (Chen et al., 2018; Kendall et al., 2018).

Therefore, we take an alternative approach and formulate this problem as multi-objective optimization with a vector-valued loss $\mathcal{L}$, as the following:

$$\min_{\substack{\boldsymbol{\theta}_g, \\ \boldsymbol{\theta}_1, \ldots, \boldsymbol{\theta}_K}} \boldsymbol{\mathcal{L}}(\mathcal{G}, \boldsymbol{\theta}_g, \boldsymbol{\theta}_1, \ldots, \boldsymbol{\theta}_K) = \min_{\substack{\boldsymbol{\theta}_g, \\ \boldsymbol{\theta}_1, \ldots, \boldsymbol{\theta}_K}} \left( \mathcal{L}_1(\mathcal{G}; \mathcal{T}_1, \boldsymbol{\theta}_g, \boldsymbol{\theta}_1), \ldots, \mathcal{L}_K(\mathcal{G}; \mathcal{T}_K, \boldsymbol{\theta}_g, \boldsymbol{\theta}_K) \right). \tag{2}$$

The focus of multi-objective optimization is approaching *Pareto optimality* (Désidéri, 2012), which in the multi-task SSL setting can be defined as follows:

**Definition 1** (Pareto Optimality). *A set of solution $(\boldsymbol{\theta}_g^\star, \boldsymbol{\theta}_1^\star, \ldots, \boldsymbol{\theta}_K^\star)$ is Pareto optimal if and only if there does not exists a set of solution that dominates $(\boldsymbol{\theta}_g^\star, \boldsymbol{\theta}_1^\star, \ldots, \boldsymbol{\theta}_K^\star)$. $(\boldsymbol{\theta}_g^\star, \boldsymbol{\theta}_1^\star, \ldots, \boldsymbol{\theta}_K^\star)$ dominates $(\hat{\boldsymbol{\theta}}_g, \hat{\boldsymbol{\theta}}_1, \ldots, \hat{\boldsymbol{\theta}}_K)$ if for every SSL task $k$, $\mathcal{L}_k(\mathcal{G}; \mathcal{T}_k, \hat{\boldsymbol{\theta}}_g, \hat{\boldsymbol{\theta}}_k) \geq \mathcal{L}_k(\mathcal{G}; \mathcal{T}_k, \boldsymbol{\theta}_g^\star, \boldsymbol{\theta}_k^\star)$ and $\boldsymbol{\mathcal{L}}(\mathcal{G}, \hat{\boldsymbol{\theta}}_g, \hat{\boldsymbol{\theta}}_1, \ldots, \hat{\boldsymbol{\theta}}_K) \neq \boldsymbol{\mathcal{L}}(\mathcal{G}, \boldsymbol{\theta}_g^\star, \boldsymbol{\theta}_1^\star, \ldots, \boldsymbol{\theta}_K^\star)$.*

In other words, if a self-supervised GNN is Pareto optimal, it is impossible to further optimize any SSL task without sacrificing the performance of at least one other SSL task. Finding the Pareto optimal model is not sensible if there exist a set of parameters that can easily fit all SSL tasks (i.e., no matter how different SSL tasks are reconciled, such a model approaches Pareto optimality where every SSL task is perfectly fitted). However, this is rarely the case, because solving all difficult pretext tasks is non-trivial for only one set of parameters. By promoting the Pareto optimlaity, PARETOGNN is enforced to learn intrinsic patterns applicable to a number of pretext tasks, which further enhances the task generalization across various downstream tasks.

## 2.3 PARETO OPTIMALITY BY MULTIPLE GRADIENT DESCENT ALGORITHM

To obtain the Pareto optimal parameters, we explore the Multiple Gradient Descent Algorithm (MGDA) (Désidéri, 2012) and adapt it to the multi-task SSL setting. Specifically, MGDA leverages the saddle-point test and theoretically proves that a solution (i.e., the combined gradient descent direction or task weight assignments in our case) that satisfies the saddle-point test gives a descent direction that improves all tasks and eventually approaches the Pareto optimality. We further elaborate descriptions of the saddle-point test for the share parameters $\boldsymbol{\theta}_g$ and task-specific parameters $\boldsymbol{\theta}_k$ in Appendix G. In our multi-task SSL scenario, the optimization problem can be formulated as:

$$\min_{\alpha_1,\dots,\alpha_K} \left\| \sum_{k=1}^{K} \alpha_k \cdot \nabla_{\boldsymbol{\theta}_g} \mathcal{L}_k(\mathcal{G};\mathcal{T}_k,\boldsymbol{\theta}_g,\boldsymbol{\theta}_k) \right\|_F, \quad \text{s.t.} \quad \sum_{k=1}^{K} \alpha_k = 1 \quad \text{and} \quad \forall k \ \alpha_k \geq 0, \quad (3)$$

where $\nabla_{\boldsymbol{\theta}_g} \mathcal{L}_k(\mathcal{G};\mathcal{T}_k,\boldsymbol{\theta}_g,\boldsymbol{\theta}_k) \in \mathbb{R}^{1 \times |\boldsymbol{\theta}_g|}$ refers to the gradients of parameters for the GNN encoder w.r.t. the $k$-th SSL task. PARETOGNN can be trained using Equation (1) with the task weights derived by the above optimization. As shown in Figure 1, optimizing the above objective is essentially finding descent direction with the minimum norm within the convex hull defined by the gradient direction of each SSL task. Hence, the solution to Equation (3) is straight-forward when $K = 2$ (i.e., only two gradient descent directions involved). If the norm of one gradient is smaller than their inner product, the solution is simply the gradient with the smaller norm (i.e., $\alpha_1 = 0$, $\alpha_2 = 1$) or vice versa). Otherwise, $\alpha_1$ can be calculated by deriving the descent direction perpendicular to the convex line with only one step, formulated as:

$$\alpha_1 = \frac{\nabla_{\boldsymbol{\theta}_g} \mathcal{L}_2(\mathcal{G};\mathcal{T}_2,\boldsymbol{\theta}_g,\boldsymbol{\theta}_2) \cdot \left( \nabla_{\boldsymbol{\theta}_g} \mathcal{L}_2(\mathcal{G};\mathcal{T}_2,\boldsymbol{\theta}_g,\boldsymbol{\theta}_2) - \nabla_{\boldsymbol{\theta}_g} \mathcal{L}_1(\mathcal{G};\mathcal{T}_1,\boldsymbol{\theta}_g,\boldsymbol{\theta}_1) \right)^{\mathsf{T}}}{\left\| \nabla_{\boldsymbol{\theta}_g} \mathcal{L}_2(\mathcal{G};\mathcal{T}_2,\boldsymbol{\theta}_g,\boldsymbol{\theta}_2) - \nabla_{\boldsymbol{\theta}_g} \mathcal{L}_1(\mathcal{G};\mathcal{T}_1,\boldsymbol{\theta}_g,\boldsymbol{\theta}_1) \right\|_F}. \quad (4)$$

When $K > 2$, we minimize the quadratic form $\boldsymbol{\alpha}(\nabla_{\boldsymbol{\theta}_g}\mathcal{L})(\nabla_{\boldsymbol{\theta}_g}\mathcal{L})^{\mathsf{T}}\boldsymbol{\alpha}^{\mathsf{T}}$, where $\nabla_{\boldsymbol{\theta}_g}\mathcal{L} \in \mathbb{R}^{K \times |\boldsymbol{\theta}_g|}$ refers to the vertically concatenated gradients w.r.t. $\boldsymbol{\theta}_g$ for all SSL tasks, and $\boldsymbol{\alpha} \in \mathbb{R}^{1 \times K}$ is the vector for task weight assignments such that $\|\boldsymbol{\alpha}\| = 1$. Inspired by Frank-Wolfe algorithm (Jaggi, 2013), we iteratively solve such a quadratic problem as a special case of Equation (4). Specifically, we first initialize every element in $\boldsymbol{\alpha}$ as $1/K$, and we increment the weight of the task (denoted as $t$) whose descent direction correlates least with the current combined descent direction (i.e., $\sum_{k=1}^{K} \alpha_k \cdot \nabla_{\boldsymbol{\theta}_g} \mathcal{L}_k(\mathcal{G};\mathcal{T}_k,\boldsymbol{\theta}_g,\boldsymbol{\theta}_k)$). The step size $\eta$ of this increment can be calculated by utilizing the idea of Equation (4), where we replace $\nabla_{\boldsymbol{\theta}_g} \mathcal{L}_2(\mathcal{G};\mathcal{T}_2,\boldsymbol{\theta}_g,\boldsymbol{\theta}_2)$ with $\sum_{k=1}^{K} \alpha_k \cdot \nabla_{\boldsymbol{\theta}_g} \mathcal{L}_k(\mathcal{G};\mathcal{T}_k,\boldsymbol{\theta}_g,\boldsymbol{\theta}_k)$ and replace $\nabla_{\boldsymbol{\theta}_g} \mathcal{L}_1(\mathcal{G};\mathcal{T}_1,\boldsymbol{\theta}_g,\boldsymbol{\theta}_1)$ with $\nabla_{\boldsymbol{\theta}_g} \mathcal{L}_t(\mathcal{G};\mathcal{T}_t,\boldsymbol{\theta}_g,\boldsymbol{\theta}_t)$. One iteration of solving this quadratic problem is formulated as:

$$\boldsymbol{\alpha} := (1 - \eta) \cdot \boldsymbol{\alpha} + \eta \cdot \mathbf{e}_t, \quad \text{s.t.} \quad \eta = \frac{\hat{\nabla}_{\boldsymbol{\theta}_g} \cdot \left( \hat{\nabla}_{\boldsymbol{\theta}_g} - \nabla_{\boldsymbol{\theta}_g} \mathcal{L}_t(\mathcal{G};\mathcal{T}_t,\boldsymbol{\theta}_g,\boldsymbol{\theta}_t) \right)^{\mathsf{T}}}{\left\| \hat{\nabla}_{\boldsymbol{\theta}_g} - \nabla_{\boldsymbol{\theta}_g} \mathcal{L}_t(\mathcal{G};\mathcal{T}_t,\boldsymbol{\theta}_g,\boldsymbol{\theta}_t) \right\|_F}, \quad (5)$$

where $t = \arg\min_r \sum_{i=1}^{K} \alpha_i \cdot \nabla_{\boldsymbol{\theta}_g} \mathcal{L}_i(\mathcal{G};\mathcal{T}_i,\boldsymbol{\theta}_g,\boldsymbol{\theta}_i) \cdot \nabla_{\boldsymbol{\theta}_g} \mathcal{L}_r(\mathcal{G};\mathcal{T}_r,\boldsymbol{\theta}_g,\boldsymbol{\theta}_r)^{\mathsf{T}}$ and $\hat{\nabla}_{\boldsymbol{\theta}_g} = \sum_{k=1}^{K} \alpha_k \cdot \nabla_{\boldsymbol{\theta}_g} \mathcal{L}_k(\mathcal{G};\mathcal{T}_k,\boldsymbol{\theta}_g,\boldsymbol{\theta}_k)$. $\mathbf{e}_t$ in the above solution refers to an one-hot vector with $t$-th element equal to 1. The optimization described in Equation (5) iterates until $\eta$ is smaller than a small constant $\xi$ or the number of iterations reaches the pre-defined number $\gamma$. Furthermore, the above task reconciliation of PARETOGNN satisfies the following theorem.

**Theorem 1.** *Assuming that $\boldsymbol{\alpha}(\nabla_{\boldsymbol{\theta}_g}\mathcal{L})(\nabla_{\boldsymbol{\theta}_g}\mathcal{L})^{\mathsf{T}}\boldsymbol{\alpha}^{\mathsf{T}}$ is $\beta$-smooth. $\boldsymbol{\alpha}$ converges to the optimal point at a rate of $\mathcal{O}(1/\gamma)$, and $\boldsymbol{\alpha}$ is at most $4\beta/(\gamma + 1)$ away from the optimal solution.*

The proof of Theorem 1 is provided in Appendix A. According to this theorem and our empirical observation, the optimization process in Equation (5) would terminate and deliver well-approximated results with $\gamma$ set to an affordable value (e.g., $\gamma = 100$).

## 3 EXPERIMENTS

### 3.1 EXPERIMENTAL SETTING

**Datasets**. We conduct comprehensive experiments on 11 real-world benchmark datasets extensively explored by the graph community. They include 8 homophilous graphs, which are `Wiki-CS`, `Pubmed`, `Amazon-Photo`, `Amazon-Computer`, `Coauthor-CS`, `Coauthor-Physics`, `ogbn-arxiv`, and `ogbn-products` (McAuley et al., 2015; Yang et al., 2016; Hu et al., 2020), as well as 3 heterophilous graphs, which are `Chameleon`, `Squirrel`, and `Actor` (Tang et al., 2009; Rozemberczki et al., 2021). Besides the graph homophily, this list of datasets covers graphs with other distinctive characteristics (i.e., from graphs with thousands of nodes to millions, and features with hundred dimensions to almost ten thousands), to fully evaluate the task generalization under different scenarios. The detailed description of these datasets can be found in Appendix C.

**Downstream Tasks and Evaluation Metrics**. We evaluate all models by four most commonly-used downstream tasks, including node classification, node clustering, link prediction, and partition prediction, whose performance is quantified by accuracy, normalized mutual information (NMI), area under the characteristic curve (AUC), and accuracy respectively, following the same evaluation protocols from previous works (Tian et al., 2014; Kipf & Welling, 2016a; Zhang & Chen, 2018).

**Evaluation Protocol**. For all downstream tasks, we follow the standard linear-evaluation protocol on graphs (Velickovic et al., 2019; Jin et al., 2022; Thakoor et al., 2022), where the parameters of the GNN encoder are frozen during the inference time and only logistic regression models (for node classification, link prediction and partition prediction) or K-Means models (for node clustering) are trained to conduct different downstream tasks. For datasets whose public splits are available (i.e., `ogbn-arxiv`, and `ogbn-products`), we utilize their given public splits for the evaluations on node classification, node clustering and partition prediction. Whereas for other datasets, we explore a random 10%/10%/80% split for the train/validation/test split, following the same setting as explored by other literature. To evaluate the performance on link prediction, for large graphs where permuting all possible edges is prohibitively expensive, we randomly sample 210,000, 30,000, and 60,000 edges for training, evaluation, and testing. And for medium-scale graphs, we explore the random split of 70%/10%/20%, following the same standard as explored by (Zhang & Chen, 2018; Zhao et al., 2022b). To prevent label leakage for link prediction, we evaluate link prediction by another identical model with testing and validation edges removed. Label for the partition prediction is induced by metis partition (Karypis & Kumar, 1998), and we explore 10 partitions for each dataset. All reported performance is averaged over 10 independent runs with different random seeds.

**Baselines**. We compare the performance of PARETOGNN with 7 state-of-the-art self-supervised GNNs, including DGI (Velickovic et al., 2019), GRACE (Zhu et al., 2020), MVGRL (Hassani & Khasahmadi, 2020), AUTOSSL (Jin et al., 2022), BGRL (Thakoor et al., 2022), CCA-SSG (Zhang et al., 2021b) and GRAPHMAE (Hou et al., 2022). These baselines are experts in at least one of the philosophies of our pretext tasks, and comparing PARETOGNN with them demonstrates the improvement brought by the multi-task self-supervised learning as well as promoting Pareto optimlaity.

**Hyper-parameters**. To ensure a fair comparison, for all models, we explore the GNN encoder with the same architecture (i.e., GCN encoder with the same layers), fix the hidden dimension of the GNN encoders, and utilize the recommended settings provided by the authors. Detailed configurations of other hyper-parameters for PARETOGNN are illustrated in Appendix D.

### 3.2 PERFORMANCE GAIN FROM THE MULTI-TASK SELF-SUPERVISED LEARNING

We conduct experiments on our pretext tasks by individually evaluating their performance as well as task generalization, as shown in Table 1. We first observe that there does not exist a single-task model that can simultaneously achieve competitive performance on every downstream task for all datasets, demonstrating that knowledge learned through a single philosophy does not suffice the strong and consistent task generalization. Models trained by a single pretext tasks alone are narrow experts (i.e., delivering satisfactory results on only few tasks or datasets) and their expertise does not translate to the strong and consistent task generalization across various downstream tasks and datasets. For instance, `TopoRec` achieves promising performance on link prediction (i.e., average rank of 3.7), but falls short on all other tasks (i.e., ranked 5.9, 5.3, and 5.9). Similarly, `MI-NSG` performs reasonably well on the partition prediction (i.e., average rank of 3.0), but underperforms on the link

Table 1: Performance and task generalization of our proposed SSL pretext tasks. `w/o Pareto` stands for combining all the objectives via vanilla weighted summation. RANK refers to the average rank among all variants given an evaluation metric. **Bold** indicates the best performance and underline indicates the runner-up, with standard deviations as subscripts.

| Method | WIKI.CS | PUBMED | AM.PHOTO | AM.COMP. | CO.CS | CO.PHY. | CHAM. | SQUIRREL | ACTOR | RANK |
|---|---|---|---|---|---|---|---|---|---|---|
| *AVERAGE PERFORMANCE* | | | | | | | | | | |
| FeatRec | 74.06 | 69.48 | 84.76 | 77.76 | 86.61 | 75.20 | 64.66 | 52.43 | 31.63 | 4.6 |
| TopoRec | 70.44 | 66.32 | 85.18 | 79.41 | 85.23 | 77.45 | 61.20 | 52.89 | 38.56 | 5.0 |
| RepDecor | 69.54 | 67.42 | 83.74 | 78.49 | 84.49 | 78.31 | 58.98 | 52.19 | 36.28 | 5.8 |
| MI-NG | 71.89 | 67.35 | 85.33 | 79.95 | 83.01 | 75.00 | 63.72 | 49.56 | 30.60 | 5.7 |
| MI-NSG | 75.59 | 69.25 | 82.99 | 80.85 | 86.02 | 80.30 | 64.69 | 53.89 | 38.22 | 3.6 |
| PARETOGNN | **76.03** | **72.48** | **86.58** | 82.57 | **87.80** | **83.35** | **65.21** | **55.31** | **40.76** | **1.0** |
| w/o Pareto | 74.64 | 69.82 | 85.82 | 82.09 | 86.54 | 82.32 | 64.37 | 54.90 | 40.12 | 2.4 |
| *NODE CLASSIFICATION (Accuracy)* | | | | | | | | | | |
| FeatRec | $81.00_{\pm0.18}$ | $82.78_{\pm0.26}$ | $93.24_{\pm0.13}$ | $89.63_{\pm0.62}$ | $92.12_{\pm0.01}$ | $\mathbf{95.49}_{\pm0.08}$ | $\mathbf{63.97}_{\pm0.08}$ | $43.94_{\pm1.13}$ | $24.41_{\pm0.61}$ | 3.7 |
| TopoRec | $80.25_{\pm0.24}$ | $81.99_{\pm0.31}$ | $92.88_{\pm0.06}$ | $87.68_{\pm0.31}$ | $90.97_{\pm0.05}$ | $94.31_{\pm0.01}$ | $63.01_{\pm0.16}$ | $40.24_{\pm0.49}$ | $25.29_{\pm0.90}$ | 5.9 |
| RepDecor | $78.11_{\pm0.26}$ | $85.41_{\pm0.14}$ | $92.21_{\pm0.06}$ | $89.05_{\pm0.24}$ | $92.53_{\pm0.03}$ | $94.31_{\pm0.26}$ | $57.46_{\pm0.49}$ | $44.37_{\pm1.52}$ | $25.11_{\pm0.14}$ | 5.2 |
| MI-NG | $80.72_{\pm0.17}$ | $81.62_{\pm0.06}$ | $93.02_{\pm0.07}$ | $89.13_{\pm0.22}$ | $89.04_{\pm0.09}$ | $93.07_{\pm0.62}$ | $62.27_{\pm0.09}$ | $41.56_{\pm0.59}$ | $25.41_{\pm0.16}$ | 5.7 |
| MI-NSG | $81.21_{\pm0.14}$ | $86.93_{\pm0.28}$ | $93.18_{\pm0.25}$ | $90.34_{\pm0.21}$ | $92.01_{\pm0.15}$ | $95.13_{\pm0.02}$ | $63.14_{\pm1.34}$ | $46.21_{\pm0.94}$ | $23.54_{\pm0.31}$ | 3.6 |
| PARETOGNN | $82.87_{\pm0.13}$ | $\mathbf{87.03}_{\pm0.25}$ | $\mathbf{93.85}_{\pm0.28}$ | $90.75_{\pm0.17}$ | $92.21_{\pm0.14}$ | $95.45_{\pm0.10}$ | $63.13_{\pm0.84}$ | $\mathbf{46.60}_{\pm1.08}$ | $\mathbf{26.62}_{\pm0.67}$ | 1.9 |
| w/o Pareto | $\mathbf{83.09}_{\pm0.11}$ | $86.81_{\pm0.10}$ | $93.69_{\pm0.04}$ | $\mathbf{90.81}_{\pm0.16}$ | $92.78_{\pm0.13}$ | $94.52_{\pm0.41}$ | $63.37_{\pm0.17}$ | $45.54_{\pm0.89}$ | $25.79_{\pm0.41}$ | 2.1 |
| *NODE CLUSTERING (NMI)* | | | | | | | | | | |
| FeatRec | $43.04_{\pm1.92}$ | $30.24_{\pm0.01}$ | $63.25_{\pm1.41}$ | $43.83_{\pm1.30}$ | $74.61_{\pm1.04}$ | $37.83_{\pm0.01}$ | $14.77_{\pm0.13}$ | $3.84_{\pm0.19}$ | $0.62_{\pm0.06}$ | 4.0 |
| TopoRec | $36.06_{\pm1.25}$ | $19.22_{\pm0.02}$ | $66.27_{\pm1.06}$ | $48.51_{\pm1.68}$ | $69.83_{\pm0.45}$ | $48.15_{\pm0.22}$ | $8.07_{\pm0.01}$ | $3.22_{\pm0.01}$ | $0.19_{\pm0.41}$ | 5.3 |
| RepDecor | $34.96_{\pm0.59}$ | $26.51_{\pm0.33}$ | $61.28_{\pm1.31}$ | $49.78_{\pm1.02}$ | $66.53_{\pm1.63}$ | $47.65_{\pm0.70}$ | $9.05_{\pm0.57}$ | $3.17_{\pm0.19}$ | $0.22_{\pm0.05}$ | 5.1 |
| MI-NG | $39.78_{\pm0.24}$ | $24.70_{\pm0.61}$ | $65.32_{\pm1.57}$ | $48.78_{\pm0.56}$ | $66.16_{\pm0.62}$ | $49.98_{\pm0.54}$ | $14.18_{\pm0.67}$ | $\mathbf{4.08}_{\pm0.04}$ | $0.91_{\pm0.04}$ | 4.1 |
| MI-NSG | $\mathbf{47.77}_{\pm0.14}$ | $24.34_{\pm0.01}$ | $55.92_{\pm1.01}$ | $49.61_{\pm0.55}$ | $74.91_{\pm0.82}$ | $56.83_{\pm0.01}$ | $\mathbf{15.33}_{\pm0.48}$ | $2.94_{\pm0.44}$ | $0.13_{\pm0.08}$ | 4.0 |
| PARETOGNN | $47.52_{\pm0.29}$ | $\mathbf{34.74}_{\pm0.06}$ | $\mathbf{68.25}_{\pm1.25}$ | $\mathbf{52.53}_{\pm0.34}$ | $\mathbf{74.94}_{\pm0.98}$ | $60.43_{\pm0.13}$ | $14.49_{\pm0.75}$ | $2.81_{\pm0.23}$ | $\mathbf{1.53}_{\pm0.04}$ | 2.1 |
| w/o Pareto | $46.12_{\pm0.25}$ | $26.18_{\pm0.41}$ | $67.22_{\pm1.12}$ | $51.85_{\pm0.24}$ | $74.04_{\pm0.33}$ | $\mathbf{60.71}_{\pm0.21}$ | $11.91_{\pm0.89}$ | $2.82_{\pm0.17}$ | $0.90_{\pm0.13}$ | 3.3 |
| *LINK PREDICTION (AUC)* | | | | | | | | | | |
| FeatRec | $95.79_{\pm0.05}$ | $93.96_{\pm0.05}$ | $95.47_{\pm0.15}$ | $90.51_{\pm0.17}$ | $96.51_{\pm0.02}$ | $95.97_{\pm0.06}$ | $94.13_{\pm0.17}$ | $89.47_{\pm0.01}$ | $67.22_{\pm0.04}$ | 4.3 |
| TopoRec | $92.69_{\pm0.25}$ | $94.17_{\pm0.94}$ | $95.13_{\pm1.25}$ | $95.89_{\pm0.12}$ | $96.43_{\pm0.37}$ | $97.98_{\pm0.01}$ | $89.35_{\pm0.49}$ | $93.91_{\pm0.33}$ | $84.03_{\pm0.45}$ | 3.7 |
| RepDecor | $93.64_{\pm0.09}$ | $87.55_{\pm0.06}$ | $94.86_{\pm0.16}$ | $86.45_{\pm0.57}$ | $94.00_{\pm0.16}$ | $96.48_{\pm0.08}$ | $87.50_{\pm0.18}$ | $86.44_{\pm0.21}$ | $71.13_{\pm0.51}$ | 6.0 |
| MI-NG | $92.48_{\pm0.08}$ | $91.48_{\pm0.17}$ | $95.33_{\pm0.05}$ | $94.19_{\pm0.04}$ | $97.83_{\pm0.11}$ | $90.18_{\pm0.15}$ | $94.26_{\pm0.07}$ | $87.26_{\pm0.04}$ | $69.16_{\pm0.53}$ | 5.0 |
| MI-NSG | $95.90_{\pm0.04}$ | $92.22_{\pm0.02}$ | $95.22_{\pm0.64}$ | $94.11_{\pm0.07}$ | $92.13_{\pm0.01}$ | $93.13_{\pm0.06}$ | $95.17_{\pm0.02}$ | $92.04_{\pm0.02}$ | $81.26_{\pm0.11}$ | 4.4 |
| PARETOGNN | $\mathbf{96.48}_{\pm0.01}$ | $\mathbf{94.58}_{\pm0.02}$ | $\mathbf{96.08}_{\pm0.08}$ | $\mathbf{97.16}_{\pm0.04}$ | $\mathbf{98.18}_{\pm0.02}$ | $\mathbf{98.33}_{\pm0.03}$ | $\mathbf{95.78}_{\pm0.05}$ | $\mathbf{96.46}_{\pm0.05}$ | $84.29_{\pm0.04}$ | 1.1 |
| w/o Pareto | $94.13_{\pm0.52}$ | $93.17_{\pm0.42}$ | $94.52_{\pm0.15}$ | $95.71_{\pm0.53}$ | $95.12_{\pm0.04}$ | $96.51_{\pm0.07}$ | $95.54_{\pm0.06}$ | $95.13_{\pm0.12}$ | $\mathbf{85.17}_{\pm0.07}$ | 3.4 |
| *PARTITION PREDICTION (Accuracy)* | | | | | | | | | | |
| FeatRec | $76.41_{\pm0.16}$ | $70.96_{\pm0.03}$ | $87.08_{\pm0.17}$ | $87.08_{\pm0.13}$ | $83.21_{\pm0.04}$ | $71.52_{\pm0.04}$ | $85.77_{\pm0.71}$ | $72.47_{\pm0.66}$ | $34.27_{\pm0.83}$ | 5.0 |
| TopoRec | $72.78_{\pm0.55}$ | $69.91_{\pm0.11}$ | $86.45_{\pm0.29}$ | $85.56_{\pm0.14}$ | $83.71_{\pm0.50}$ | $69.38_{\pm0.45}$ | $84.39_{\pm0.58}$ | $74.20_{\pm0.72}$ | $44.74_{\pm0.11}$ | 5.9 |
| RepDecor | $71.47_{\pm0.05}$ | $70.23_{\pm0.14}$ | $86.59_{\pm0.09}$ | $88.67_{\pm0.12}$ | $84.91_{\pm0.10}$ | $74.80_{\pm0.66}$ | $81.86_{\pm0.52}$ | $74.77_{\pm0.51}$ | $48.67_{\pm0.54}$ | 4.7 |
| MI-NG | $74.57_{\pm0.39}$ | $71.60_{\pm0.30}$ | $87.67_{\pm0.18}$ | $87.70_{\pm0.14}$ | $79.01_{\pm0.49}$ | $66.76_{\pm0.41}$ | $84.17_{\pm0.74}$ | $65.34_{\pm0.16}$ | $\mathbf{26.92}_{\pm0.59}$ | 5.7 |
| MI-NSG | $\mathbf{77.49}_{\pm0.39}$ | $73.50_{\pm0.16}$ | $87.62_{\pm0.07}$ | $89.34_{\pm0.09}$ | $85.01_{\pm0.23}$ | $76.13_{\pm0.12}$ | $85.11_{\pm0.76}$ | $74.37_{\pm0.25}$ | $47.96_{\pm0.54}$ | 3.0 |
| PARETOGNN | $77.23_{\pm0.36}$ | $\mathbf{73.57}_{\pm0.23}$ | $\mathbf{88.13}_{\pm0.39}$ | $89.84_{\pm0.06}$ | $\mathbf{85.89}_{\pm0.03}$ | $\mathbf{79.19}_{\pm0.20}$ | $\mathbf{87.43}_{\pm1.04}$ | $75.39_{\pm0.65}$ | $\mathbf{50.61}_{\pm0.75}$ | 1.3 |
| w/o Pareto | $75.22_{\pm0.11}$ | $73.12_{\pm0.51}$ | $87.84_{\pm0.02}$ | $\mathbf{89.97}_{\pm0.41}$ | $84.21_{\pm0.44}$ | $77.54_{\pm0.33}$ | $86.66_{\pm0.83}$ | $\mathbf{76.09}_{\pm0.41}$ | $48.61_{\pm0.16}$ | 2.4 |

prediction task (i.e., average rank of 4.4). However, comparing them with the model trained by combining all pretext tasks through the weighted summation (i.e., `w/o Pareto`), we observe that the latter achieves both stronger task generalization and better single-task performance. The model `w/o Pareto` achieves an average rank of 2.4 on the average performance, which is 1.2 ranks higher than the best single-task model. This phenomenon indicates that multi-task self-supervised GNNs indeed enable stronger task generalization. Multiple objectives regularize the learning model against extracting redundant information so that the model learns multiple complementary views of the given graphs. Besides, multi-task training also improves the performance of the single downstream task by 1.5, 0.7, 0.3, and 0.6 ranks, respectively.

In some cases (e.g., node clustering on PUBMED and CHAMELEON, or link prediction on WIKI.CS and CO.CS), we observe large performance margins between the best-performing single-task models and the vanilla multi-task model `w/o Pareto`, indicating that there exist potential conflicts between different SSL tasks. PARETOGNN further mitigates these performance margins by promoting Pareto optimality, which enforces the learning model to capture intrinsic patterns applicable to a number of pretext tasks while minimizing potential conflicts. As shown in Table 1, PARETOGNN is the top-ranked at both average metric and other individual downstream tasks, demonstrating the strong task generalization as well as the promising single-task performance. Specifically, PARETOGNN achieves an outstanding average rank of 1.0 on the average performance of the four downstream tasks. As for the performance on individual tasks, PARETOGNN achieves an average rank of 1.7, 1.6, 1.0, and 1.2 on the four individual downstream tasks, outperforming the corresponding best baselines by 0.2, 1.2, 1.3, and 1.1 respectively. This phenomenon demonstrates that promoting Pareto optimality not only helps improve the task generalization across various tasks and datasets but also enhances the single-task performance for the multi-task self-supervised learning.

Table 2: Performance and task generalization of PARETOGNN as well as the state-of-the-art unsupervised baselines. OOM stands for out-of-memory on a RTX3090 GPU with 24 GB memory.

| Method | Wiki.CS | PubMed | Am.Photo | Am.Comp. | Co.CS | Co.Phy. | Cham. | Squirrel | Actor | Rank |
|---|---|---|---|---|---|---|---|---|---|---|
| | | | | | AVERAGE PERFORMANCE | | | | | |
| DGI | 72.50 | 59.84 | 83.67 | 77.91 | 85.67 | 79.86 | 61.34 | 50.23 | 33.77 | 4.4 |
| GRACE | 71.01 | 65.04 | 80.87 | 76.06 | 87.16 | OOM | 62.12 | 51.02 | 32.59 | 4.8 |
| MVGRL | 68.59 | 63.23 | 82.49 | 67.80 | 82.69 | 75.72 | 59.77 | 45.01 | 31.22 | 7.1 |
| AUTOSSL | 71.72 | 65.90 | 83.96 | 77.02 | 85.88 | 80.03 | 60.87 | 49.76 | 31.33 | 4.4 |
| BGRL | 74.68 | 67.15 | 80.14 | 79.06 | 87.26 | 81.42 | 60.20 | 49.14 | 32.33 | 3.8 |
| CCA-SSG | 73.44 | 64.12 | 83.62 | 78.71 | 83.65 | 79.77 | 62.80 | 51.63 | 35.90 | 3.7 |
| GRAPHMAE | 67.83 | 62.15 | 76.94 | 72.18 | 86.80 | 77.48 | 58.27 | 45.22 | 31.49 | 6.8 |
| PARETOGNN | **76.03** | **72.48** | **86.58** | **82.57** | **87.80** | **83.35** | **65.21** | **55.31** | **40.76** | **1.0** |
| | | | | | NODE CLASSIFICATION (Accuracy) | | | | | |
| DGI | $75.53_{\pm0.09}$ | $83.52_{\pm0.52}$ | $91.61_{\pm0.17}$ | $83.59_{\pm0.22}$ | $92.15_{\pm0.25}$ | $94.51_{\pm0.27}$ | $61.00_{\pm1.68}$ | $40.54_{\pm0.62}$ | $25.19_{\pm0.52}$ | 5.9 |
| GRACE | $80.14_{\pm0.09}$ | $86.06_{\pm0.35}$ | $92.78_{\pm0.49}$ | $89.53_{\pm0.34}$ | $91.12_{\pm0.25}$ | OOM | $58.19_{\pm0.53}$ | $41.36_{\pm0.47}$ | $24.63_{\pm0.39}$ | 5.0 |
| MVGRL | $79.11_{\pm0.17}$ | $83.62_{\pm0.10}$ | $92.48_{\pm0.09}$ | $82.78_{\pm0.15}$ | $90.33_{\pm0.17}$ | $91.05_{\pm0.14}$ | $49.87_{\pm0.14}$ | $39.81_{\pm0.40}$ | $28.01_{\pm0.41}$ | 6.4 |
| AUTOSSL | $79.55_{\pm0.23}$ | $86.26_{\pm0.20}$ | $92.71_{\pm0.43}$ | $88.76_{\pm0.52}$ | $92.17_{\pm0.17}$ | $95.13_{\pm0.43}$ | $58.94_{\pm0.94}$ | $40.63_{\pm0.62}$ | $24.54_{\pm0.28}$ | 4.7 |
| BGRL | $82.58_{\pm0.27}$ | $86.03_{\pm0.17}$ | $93.17_{\pm0.23}$ | $90.15_{\pm0.18}$ | $91.77_{\pm0.47}$ | $95.73_{\pm0.23}$ | $56.05_{\pm1.06}$ | $41.64_{\pm0.79}$ | $24.03_{\pm0.78}$ | 4.1 |
| CCA-SSG | $82.48_{\pm0.35}$ | $86.36_{\pm0.35}$ | $93.29_{\pm0.49}$ | $89.58_{\pm0.70}$ | $94.24_{\pm0.17}$ | $95.63_{\pm0.09}$ | $57.39_{\pm1.38}$ | $42.22_{\pm0.94}$ | $26.35_{\pm0.35}$ | 2.8 |
| GRAPHMAE | $77.12_{\pm0.30}$ | $83.91_{\pm0.26}$ | $90.71_{\pm0.40}$ | $79.44_{\pm0.48}$ | $93.13_{\pm0.15}$ | $95.79_{\pm0.06}$ | $55.50_{\pm0.82}$ | $35.87_{\pm0.61}$ | $28.97_{\pm0.27}$ | 5.3 |
| PARETOGNN | $82.87_{\pm0.13}$ | $87.03_{\pm0.25}$ | $93.85_{\pm0.28}$ | $90.75_{\pm0.17}$ | $92.21_{\pm0.14}$ | $95.45_{\pm0.10}$ | $63.13_{\pm0.84}$ | $46.60_{\pm1.08}$ | $26.62_{\pm0.67}$ | **1.8** |
| | | | | | NODE CLUSTERING (NMI) | | | | | |
| DGI | $44.35_{\pm0.12}$ | $9.68_{\pm0.31}$ | $60.31_{\pm0.23}$ | $47.76_{\pm0.02}$ | $72.88_{\pm0.21}$ | $58.76_{\pm0.43}$ | $6.99_{\pm1.74}$ | $2.16_{\pm0.14}$ | $1.49_{\pm0.05}$ | 5.1 |
| GRACE | $40.40_{\pm0.10}$ | $25.64_{\pm0.24}$ | $55.20_{\pm0.70}$ | $41.77_{\pm0.32}$ | $76.61_{\pm0.26}$ | OOM | $11.73_{\pm1.01}$ | $2.53_{\pm0.31}$ | $1.09_{\pm0.22}$ | 5.3 |
| MVGRL | $36.20_{\pm0.29}$ | $24.87_{\pm0.13}$ | $56.36_{\pm0.08}$ | $31.27_{\pm0.29}$ | $73.34_{\pm0.01}$ | $58.27_{\pm0.01}$ | $18.57_{\pm0.26}$ | $4.40_{\pm0.21}$ | $2.63_{\pm0.06}$ | 4.3 |
| AUTOSSL | $36.99_{\pm0.21}$ | $28.99_{\pm0.26}$ | $64.06_{\pm0.65}$ | $41.85_{\pm0.36}$ | $74.04_{\pm0.22}$ | $55.23_{\pm0.18}$ | $9.67_{\pm1.21}$ | $2.11_{\pm0.12}$ | $1.35_{\pm0.11}$ | 5.0 |
| BGRL | $44.95_{\pm0.32}$ | $26.38_{\pm0.36}$ | $50.56_{\pm0.24}$ | $44.04_{\pm0.36}$ | $74.06_{\pm0.51}$ | $61.04_{\pm0.11}$ | $11.29_{\pm1.45}$ | $2.28_{\pm0.53}$ | $1.32_{\pm0.07}$ | 4.3 |
| CCA-SSG | $44.17_{\pm0.04}$ | $27.15_{\pm1.56}$ | $64.06_{\pm0.03}$ | $49.78_{\pm0.01}$ | $67.14_{\pm0.49}$ | $54.33_{\pm2.69}$ | $12.17_{\pm0.79}$ | $3.02_{\pm0.04}$ | $1.01_{\pm0.02}$ | 4.3 |
| GRAPHMAE | $35.73_{\pm0.04}$ | $19.00_{\pm0.11}$ | $51.42_{\pm0.46}$ | $43.51_{\pm0.16}$ | $76.18_{\pm0.59}$ | $46.90_{\pm0.07}$ | $8.66_{\pm0.47}$ | $3.70_{\pm0.32}$ | $1.24_{\pm0.05}$ | 5.7 |
| PARETOGNN | $47.52_{\pm0.29}$ | $34.74_{\pm0.06}$ | $68.25_{\pm1.25}$ | $52.53_{\pm0.34}$ | $74.94_{\pm0.98}$ | $60.43_{\pm0.13}$ | $14.49_{\pm0.23}$ | $2.81_{\pm0.23}$ | $1.53_{\pm0.04}$ | **1.9** |
| | | | | | LINK PREDICTION (AUC) | | | | | |
| DGI | $94.36_{\pm0.04}$ | $78.59_{\pm0.58}$ | $94.24_{\pm0.16}$ | $90.37_{\pm0.03}$ | $93.46_{\pm0.12}$ | $88.82_{\pm0.05}$ | $91.64_{\pm0.62}$ | $92.45_{\pm0.01}$ | $72.01_{\pm0.41}$ | 5.0 |
| GRACE | $92.32_{\pm0.05}$ | $87.44_{\pm1.03}$ | $91.82_{\pm0.12}$ | $88.58_{\pm0.08}$ | $97.00_{\pm0.02}$ | OOM | $93.81_{\pm0.47}$ | $93.57_{\pm0.06}$ | $82.31_{\pm0.17}$ | 4.6 |
| MVGRL | $94.34_{\pm0.20}$ | $90.31_{\pm0.02}$ | $93.86_{\pm0.25}$ | $75.15_{\pm0.34}$ | $92.44_{\pm0.04}$ | $87.54_{\pm0.08}$ | $94.50_{\pm0.44}$ | $88.81_{\pm0.33}$ | $77.93_{\pm0.07}$ | 5.0 |
| AUTOSSL | $93.86_{\pm0.02}$ | $86.84_{\pm1.30}$ | $95.57_{\pm0.13}$ | $93.99_{\pm0.03}$ | $95.71_{\pm0.15}$ | $95.93_{\pm0.07}$ | $90.05_{\pm0.78}$ | $93.84_{\pm0.19}$ | $70.51_{\pm0.29}$ | 4.3 |
| BGRL | $94.31_{\pm0.06}$ | $94.35_{\pm0.02}$ | $93.33_{\pm0.06}$ | $93.59_{\pm0.05}$ | $97.37_{\pm0.04}$ | $93.38_{\pm0.07}$ | $92.24_{\pm0.62}$ | $83.60_{\pm0.10}$ | $76.91_{\pm0.10}$ | 4.4 |
| CCA-SSG | $90.92_{\pm0.33}$ | $73.53_{\pm1.50}$ | $89.47_{\pm0.13}$ | $86.72_{\pm0.13}$ | $91.77_{\pm0.03}$ | $93.64_{\pm0.02}$ | $95.43_{\pm0.31}$ | $87.30_{\pm0.25}$ | $78.14_{\pm0.09}$ | 5.7 |
| GRAPHMAE | $89.47_{\pm0.02}$ | $86.24_{\pm0.19}$ | $83.33_{\pm0.07}$ | $84.65_{\pm1.08}$ | $96.48_{\pm0.17}$ | $96.57_{\pm0.08}$ | $89.40_{\pm1.48}$ | $86.48_{\pm0.14}$ | $80.65_{\pm0.28}$ | 5.9 |
| PARETOGNN | $96.48_{\pm0.01}$ | $94.58_{\pm0.02}$ | $96.08_{\pm0.08}$ | $97.16_{\pm0.04}$ | $98.18_{\pm0.01}$ | $98.33_{\pm0.12}$ | $95.78_{\pm0.05}$ | $96.46_{\pm0.05}$ | $84.29_{\pm0.04}$ | **1.0** |
| | | | | | PARTITION PREDICTION (Accuracy) | | | | | |
| DGI | $75.75_{\pm0.19}$ | $67.58_{\pm0.01}$ | $\mathbf{88.50_{\pm0.20}}$ | $89.91_{\pm0.07}$ | $84.19_{\pm0.66}$ | $77.34_{\pm0.01}$ | $85.75_{\pm0.14}$ | $65.77_{\pm0.02}$ | $36.40_{\pm0.02}$ | 2.9 |
| GRACE | $71.19_{\pm0.13}$ | $61.01_{\pm1.57}$ | $83.66_{\pm0.83}$ | $84.36_{\pm0.21}$ | $83.89_{\pm0.82}$ | OOM | $84.76_{\pm2.03}$ | $66.64_{\pm0.12}$ | $22.32_{\pm0.05}$ | 5.4 |
| MVGRL | $64.73_{\pm0.25}$ | $54.11_{\pm0.06}$ | $87.25_{\pm0.44}$ | $81.99_{\pm0.04}$ | $74.66_{\pm0.19}$ | $66.03_{\pm0.35}$ | $76.12_{\pm1.03}$ | $47.03_{\pm0.81}$ | $16.31_{\pm0.04}$ | 7.2 |
| AUTOSSL | $76.47_{\pm0.07}$ | $61.52_{\pm0.01}$ | $83.51_{\pm0.05}$ | $83.46_{\pm0.79}$ | $81.59_{\pm0.80}$ | $73.85_{\pm0.04}$ | $84.83_{\pm0.94}$ | $62.45_{\pm0.02}$ | $28.90_{\pm0.04}$ | 5.0 |
| BGRL | $76.87_{\pm0.03}$ | $61.84_{\pm0.02}$ | $83.51_{\pm0.05}$ | $88.45_{\pm0.38}$ | $85.84_{\pm1.29}$ | $75.53_{\pm0.01}$ | $81.21_{\pm0.99}$ | $69.04_{\pm0.39}$ | $27.07_{\pm0.04}$ | 3.9 |
| CCA-SSG | $76.18_{\pm0.46}$ | $69.46_{\pm0.26}$ | $87.67_{\pm0.31}$ | $88.77_{\pm0.19}$ | $81.44_{\pm0.23}$ | $75.49_{\pm0.15}$ | $86.22_{\pm0.66}$ | $73.97_{\pm0.22}$ | $38.11_{\pm0.29}$ | 3.1 |
| GRAPHMAE | $69.01_{\pm0.27}$ | $59.46_{\pm0.08}$ | $82.31_{\pm0.13}$ | $81.11_{\pm1.57}$ | $81.39_{\pm0.20}$ | $70.67_{\pm0.16}$ | $79.50_{\pm1.26}$ | $54.82_{\pm0.67}$ | $15.09_{\pm0.01}$ | 7.2 |
| PARETOGNN | $77.23_{\pm0.36}$ | $73.57_{\pm0.23}$ | $88.13_{\pm0.39}$ | $89.84_{\pm0.06}$ | $85.89_{\pm0.33}$ | $79.19_{\pm0.20}$ | $87.43_{\pm1.05}$ | $75.39_{\pm0.65}$ | $50.61_{\pm0.75}$ | **1.2** |

## 3.3 PERFORMANCE COMPARED WITH OTHER UNSUPERVISED BASELINES

We compare PARETOGNN with 7 state-of-the-art SSL frameworks for graphs, as shown in Table 2. Similar to our previous observations, there does not exist any SSL baseline that can simultaneously achieve competitive performance on every downstream task for all datasets. Specifically, from the perspective of individual tasks, CCA-SSG performs well on the node classification but underperforms on the link prediction and the partition prediction. Besides, DGI works well on the partition prediction but performs not as good on other tasks. Nevertheless, from the perspective of datasets, we can observe that AUTOSSL has good performance on homohpilous datasets but this phenomenon does not hold for heterophilous ones, demonstrating that the assumption of reconciling tasks by promoting graph homophily is not applicable to all graphs. PARETOGNN achieves a competitive average rank of 1.0 on the average performance, significantly outperforming the runner-ups by 2.7 ranks. Moreover, PARETOGNN achieves an average rank of 1.8, 1.9, 1.0, and 1.2 on the four individual tasks, outrunning the best-performing baseline by 0.9, 2.3, 3.3, and 1.6 respectively, which further proves the strong task generalization and single-task performance of PARETOGNN.

## 3.4 PERFORMANCE WHEN SCALING TO LARGER DIMENSIONS

To evaluate the scalability of PARETOGNN, we conduct experiments from two perspectives: the graph and model dimensions. We expect our proposal to retain its strong task generalization when applied to large graphs. And we should expect even stronger performance when the model dimension scales up, compared with other single-task frameworks with larger dimensions as well, since multi-task SSL settings enable the model capable of learning more. The results are shown below.

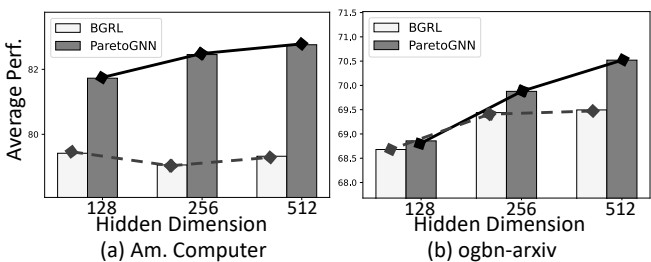

Figure 2: Task generalization (i.e., average performance) over `Amazon-Computer` and `ogbn-arxiv` for PARETOGNN and BGRL w.r.t. hidden dimensions of the graph encoder.

Table 3: Task generalization on large graphs. (*: Graphs are sampled by GRAPHSAINT (Zeng et al., 2019) matching the memory of others due to OOM.) Individual tasks are reported in Appendix F.

| Method | ARXIV | PRODUCTS | RANK |
|---|---|---|---|
| | AVERAGE PERFORMANCE | | |
| DGI | 69.75 | 75.48* | 3.5 |
| GRACE | 68.82* | 75.58* | 4.0 |
| AUTOSSL | 68.02 | OOM | 7.0 |
| BGRL | 70.04 | 75.19* | 3.5 |
| CCA-SSG | 69.62 | 72.92* | 5.5 |
| GRAPHMAE | 71.15 | 74.89* | 3.5 |
| PARETOGNN | **71.39** | **75.98*** | **1.0** |

From Figure 2, we observe that the task generalization of PARETOGNN is proportional to the model dimension, indicating that PARETOGNN is capable of learning more, compared with single-task models like BGRL, whose performance gets saturated with less-parameterized models (e.g., 128 for AM.COMP. and 256 for ARXIV). Moreover, from Table 3, we notice that the graph dimension is not a limiting factor for the strong task generalization of our proposal. Specifically, PARETOGNN outperforms the runner-ups by 2.5 on the rank of the average performance.

## 4 RELATED WORKS

**Graph Neural Networks**. Graph neural networks (GNNs) are powerful learning frameworks to extract representative information from graphs (Kipf & Welling, 2016a; Veličković et al., 2017; Hamilton et al., 2017; Xu et al., 2018b; Klicpera et al., 2019; Xu et al., 2018a; Fan et al., 2022; Zhang et al., 2019). They aim at mapping the input nodes into low-dimensional vectors, which can be further utilized to conduct either graph-level or node-level tasks. Most GNNs explore layer-wise message passing scheme (Gilmer et al., 2017; Ju et al., 2022a), where a node iteratively extracts information from its first-order neighbors. They are applied in many real-world applications, such as predictive user behavior modeling (Zhang et al., 2021c; Wen et al., 2022), molecular property prediction (Zhang et al., 2021e; Guo et al., 2021), and question answering (Ju et al., 2022b).

**Self-supervised Learning for GNNs**. For node-level tasks, current state-of-the-art graph SSL frameworks are mostly introduced according to a single pretext task with a single philosophy (You et al., 2020b), such as mutual information maximization (Velickovic et al., 2019; Zhu et al., 2020; Hassani & Khasahmadi, 2020; Thakoor et al., 2022), whitening decorrelation (Zhang et al., 2021b), and generative reconstruction (Kipf & Welling, 2016b; Hou et al., 2022). Whereas for graph-level tasks, previous works explore mutual information maximization to encourage different augmented views of the same graphs sharing similar representations (You et al., 2020a; Xu et al., 2021; You et al., 2021; Li et al., 2022; Zhao et al., 2022a).

**Multi-Task Self-supervised Learning**. Multi-task SSL is broadly explored in computer vision (Lu et al., 2020; Doersch & Zisserman, 2017; Ren & Lee, 2018; Yu et al., 2020; Ni et al., 2021) and natural language processing (Wang et al., 2018; Radford et al., 2019; Sanh et al., 2021; Ravanelli et al., 2020) fields. For the graph community, AutoSSL (Jin et al., 2022) explores a multi-task setting where tasks are reconciled in a way that promotes graph homophily (Zhao et al., 2021b). Besides, Hu et al. (2019) focused on the graph-level tasks and pre-trains GNNs in separate stages. In these frameworks, tasks are reconciled according to weighted summation or pre-defined heuristics.

## 5 CONCLUSION

We study the problem of task generalization for SSL-based GNNs in a more rigorous setting and demonstrate that their promising performance on one task or two usually does not translate into good task generalization across various downstream tasks and datasets. In light of this, we propose PARETOGNN to enhance the task generalization by multi-task self-supervised learning. Specifically, PARETOGNN is self-supervised by manifold pretext tasks observing multiple philosophies, which are reconciled by a multiple-gradient descent algorithm promoting Pareto optimality. Through our extensive experiments, we show that multi-task SSL indeed enhances the task generalization. Aided by our proposed task reconciliation, PARETOGNN further enlarges the margin by actively learning from multiple tasks while minimizing potential conflicts. Compared with 7 state-of-the-art SSL-based GNNs, PARETOGNN is top-ranked on the average performance. Besides stronger task generalization, PARETOGNN achieves better single-task performance, demonstrating that disjoint yet complementary knowledge from different philosophies is learned through the multi-task SSL.

## ACKNOWLEDGMENTS

This work is partially supported by the NSF under grants IIS-2209814, IIS-2203262, IIS-2214376, IIS-2217239, OAC-2218762, CNS-2203261, CNS-2122631, CMMI-2146076, and the NIJ 2018-75-CX-0032. Any opinions, findings, and conclusions or recommendations expressed in this material are those of the authors and do not necessarily reflect the views of any funding agencies.

## ETHICS STATEMENT

We observe no ethical concern entailed by our proposal, but we note that both ethical or unethical applications based on graphs may benefit from the stronger task generalization of our work. Care should be taken to ensure socially positive and beneficial results of machine learning algorithms.

## REPRODUCIBILITY STATEMENT

Our code is publicly available at `https://github.com/jumxglhf/ParetoGNN`. The hyper-parameters and other variables required to reproduce our experiments are described in Appendix D.

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

## A    PROOF TO THEOREM 1

Here we re-state Theorem 1 before diving into its proof:

**Theorem 1.** *Assuming that* $\boldsymbol{\alpha}(\nabla_{\boldsymbol{\theta}_g}\mathcal{L})(\nabla_{\boldsymbol{\theta}_g}\mathcal{L})^\intercal\boldsymbol{\alpha}^\intercal$ *is* $\beta$-*smooth.* $\boldsymbol{\alpha}$ *converges to the optimal point at a rate of* $\mathcal{O}(1/\gamma)$, *and* $\boldsymbol{\alpha}$ *is at most* $4\beta/(\gamma+1)$ *away from the optimal solution.*

*Proof.* Let $\phi(\alpha)$ denotes $\boldsymbol{\alpha}(\nabla_{\boldsymbol{\theta}_g}\mathcal{L})(\nabla_{\boldsymbol{\theta}_g}\mathcal{L})^\intercal\boldsymbol{\alpha}^\intercal$. If $\phi(\cdot)$ is $\beta$-smooth, we have:

$$||\nabla\phi(\boldsymbol{\alpha}) - \nabla\phi(\boldsymbol{\alpha}')|| \le \beta \cdot ||\boldsymbol{\alpha} - \boldsymbol{\alpha}'||, \tag{6}$$

from which we can also derive a quadratic upper-bound:

$$\phi(\boldsymbol{\alpha}') \le \phi(\boldsymbol{\alpha}) + \nabla\phi(\boldsymbol{\alpha})^\intercal \cdot (\boldsymbol{\alpha}' - \boldsymbol{\alpha}) + \frac{\beta}{2} \cdot ||\boldsymbol{\alpha}' - \boldsymbol{\alpha}||^2, \tag{7}$$

where $\boldsymbol{\alpha}'$ refers to the weight combination after one iteration from $\boldsymbol{\alpha}$.

Combining Equations (5) and (7), we have:

$$
\begin{aligned}
\phi(\boldsymbol{\alpha}') - \phi(\boldsymbol{\alpha}) &\le \nabla\phi(\boldsymbol{\alpha})^\intercal \cdot (\boldsymbol{\alpha}' - \boldsymbol{\alpha}) + \frac{\beta}{2} \cdot ||\boldsymbol{\alpha}' - \boldsymbol{\alpha}||^2, \\
&\le \eta \cdot \nabla\phi(\boldsymbol{\alpha})^\intercal \cdot (\mathbf{e}_t - \boldsymbol{\alpha}) + \frac{\beta}{2}\eta^2 \cdot R^2,
\end{aligned}
\tag{8}
$$

where $R = \text{SUP}_{\boldsymbol{\alpha}_1,\boldsymbol{\alpha}_2\in\mathcal{X}}(||\boldsymbol{\alpha}_1-\boldsymbol{\alpha}_2||)$ refers to the diameter of the domain of $\boldsymbol{\alpha}(\nabla_{\boldsymbol{\theta}_g}\mathcal{L})(\nabla_{\boldsymbol{\theta}_g}\mathcal{L})^\intercal\boldsymbol{\alpha}^\intercal$, and $\text{SUP}(\cdot)$ is the supremum operation. In our case, $R = \sqrt{2}$ since $||\boldsymbol{\alpha}|| = 1$. The derivation above is valid because $\boldsymbol{\alpha}' = (1 - \eta) \cdot \boldsymbol{\alpha} + \eta \cdot \mathbf{e}_t$, as shown in Equation (5).

Since $t = \arg\min_r \sum_{i=1}^{K} \alpha_i \cdot \nabla_{\boldsymbol{\theta}_g}\mathcal{L}_i(\mathcal{G};\mathcal{T}_i,\boldsymbol{\theta}_g,\boldsymbol{\theta}_i) \cdot \nabla_{\boldsymbol{\theta}_g}\mathcal{L}_r(\mathcal{G};\mathcal{T}_r,\boldsymbol{\theta}_g,\boldsymbol{\theta}_r)^\intercal$, we have:

$$
\begin{aligned}
\phi(\boldsymbol{\alpha}') - \phi(\boldsymbol{\alpha}) &\le \eta \cdot \nabla\phi(\boldsymbol{\alpha})^\intercal \cdot (\boldsymbol{\alpha}^* - \boldsymbol{\alpha}) + \beta \cdot \eta^2, \\
&\le \eta \cdot \big(\phi(\boldsymbol{\alpha}^*) - \phi(\boldsymbol{\alpha})\big) + \beta \cdot \eta^2,
\end{aligned}
\tag{9}
$$

where $\boldsymbol{\alpha}^*$ is the unknown optimal solution. By rearranging terms in the above inequality, we have:

$$\phi(\boldsymbol{\alpha}') - \phi(\boldsymbol{\alpha}^*) \le (1 - \eta) \cdot \big(\phi(\boldsymbol{\alpha}) - \phi(\boldsymbol{\alpha}^*)\big) + \beta \cdot \eta^2. \tag{10}$$

The left hand side of the above equation is the distance between the updated solution and the optimal solution (we denote this distance as $\delta' = \phi(\boldsymbol{\alpha}') - \phi(\boldsymbol{\alpha}^*)$). And on the right hand side, inside the parenthesis we have the distance between the previous solution and the optimal solution (denoting this distance as $\delta = \phi(\boldsymbol{\alpha}) - \phi(\boldsymbol{\alpha}^*)$) as:

$$\delta' \le (1 - \eta) \cdot \delta + \beta \cdot \eta^2, \tag{11}$$

where in our case, $\eta = \frac{\hat{\nabla}_{\boldsymbol{\theta}_g} \cdot \left(\hat{\nabla}_{\boldsymbol{\theta}_g} - \nabla_{\boldsymbol{\theta}_g}\mathcal{L}_t(\mathcal{G};\mathcal{T}_t,\boldsymbol{\theta}_g,\boldsymbol{\theta}_t)\right)^\intercal}{\left|\left|\hat{\nabla}_{\boldsymbol{\theta}_g} - \nabla_{\boldsymbol{\theta}_g}\mathcal{L}_t(\mathcal{G};\mathcal{T}_t,\boldsymbol{\theta}_g,\boldsymbol{\theta}_t)\right|\right|_F}$. In this proof, we assume $\eta = \frac{2}{\gamma_t+1}$, where $\gamma_t$ refers to the index of the current iteration. Such an assumption is a relaxed version of our formulation upon $\eta$, hence any proof that holds for this assumption also holds for our case.

Next, we show $\delta_\gamma \le \frac{4\beta}{\gamma+1}$ in Theorem 1 through proof-by-induction:

It's straight-forward and easy to validate that our derivation stands for the base case $\gamma = 2$, where $\delta_2 \le \frac{4\beta}{3}$. Here we show that it also holds for $\gamma + 1$:

$$
\begin{aligned}
\delta_{\gamma+1} &\le (1 - \eta) \cdot \delta_\gamma + \beta \cdot \eta^2, \\
&\le (1 - \frac{2}{\gamma + 1}) \cdot \frac{4\beta}{\gamma + 1} + \beta \cdot (\frac{2}{\gamma + 1})^2, \\
&= \frac{\gamma - 1}{\gamma + 1} \cdot \frac{4\beta}{\gamma + 1} + \frac{4\beta}{(\gamma + 1)^2}, \\
&= \frac{4\beta}{\gamma + 1} \cdot \frac{\gamma}{\gamma + 1}, \\
&\le \frac{4\beta}{\gamma + 1}
\end{aligned}
\tag{12}
$$

$\square$

## B DESCRIPTION OF THE PRETEXT TASKS

In this section, we demonstrate the design of our proposed five pretext tasks, including two based on generative reconstruction (i.e., `FeatRec` and `TopoRec`), one based on whitening decorrelation (i.e., `RepDecor`) and two based on mutual information maximization (i.e., `MI-NG` and `MI-NSG`).

We first explain the operation of graph convolution as proposed by Kipf & Welling (2016a). The key mechanism of graph convolution is layer-wise message passing where a node iteratively extracts information from its first-order neighbors and information from milti-hop neighbors can be captured through stacked convolution layers. Specifically, at $l$-th layer, this process is formulated as follows:

$$\mathbf{H}^{l+1} = \sigma(\mathbf{A} \cdot \mathbf{H}^l \cdot \mathbf{W}^l), \tag{13}$$

where $\mathbf{H}^0 = \mathbf{X}$, $\mathbf{A} \in \{0,1\}^{N \times N}$ is the adjacency matrix of the input graph, $\sigma(\cdot)$ refers to the non-linear activation function, $\mathbf{W}^l \in \mathbb{R}^{d^l \times d^{l+1}}$ refers to the learnable parameters of $l$-th layer, and $d^l$ and $d^{l+1}$ are the hidden dimensions at these two consecutive layers respectively. The graph encoder $f_g(\cdot; \boldsymbol{\theta}_g) : \mathcal{G} \to \mathbb{R}^{N \times d}$ of PARETOGNN is constructed by stacked graph convolution layers as:

$$f_g(\mathcal{G}; \boldsymbol{\theta}_g) = \mathbf{H}^L = \sigma(\mathbf{A} \cdot \mathbf{H}^{L-1} \cdot \mathbf{W}^{L-1}), \tag{14}$$

where $L$ stands for the number of layers in the encoder of PARETOGNN and $\boldsymbol{\theta}_g = \{\mathbf{W}^l\}_{l=0}^{L-1}$.

As briefly described in Section 2.1, we regard the full graph $\mathcal{G}$ as the data source; and for each task, PARETOGNN is self-supervised by sub-graphs sampled from $\mathcal{G}$, followed by task-specific augmentations (i.e., $\mathcal{T}_t(\cdot)$). The graph sampling strategy is fairly straightforward. For the pretext task $t$, we select the sub-graph constituted by nodes within $k_t$ hops of $N_t$ randomly selected seed nodes, where $k_t$ and $N_t$ are two task-specific hyper-parameters. The graph augmentation operations we explore include feature masking, edge dropping, and node dropping. For the simplicity of denotation, we unify the operation of graph sampling and graph augmentation together as $\mathcal{T}_t(\cdot)$. Task-specific hyper-parameters for graph augmentation and sub-graph sampling are covered in Appendix D.

### B.1 GENERATIVE RECONSTRUCTION

**Feature reconstruction**, denoted as `FeatRec`, utilizes the high-level idea from Zhang et al. (2021d), proving that topological information can be referred purely from the node features. Hence to utilize such inductive bias, following the implementation of GraphMAE (Hou et al., 2022), we mask the node features and forward the masked graphs through $f_g(\cdot; \boldsymbol{\theta}_g)$. Then we re-mask the previous masked nodes and feed the resulted graph to a convolution-based decoder, formulated as:

$$\hat{\mathbf{X}}' = \mathbf{A}' \cdot f_g(\mathcal{G}'; \boldsymbol{\theta}_g) \odot \mathbf{M} \cdot \mathbf{W}^{\text{Dec}}, \tag{15}$$

where $\odot$ refers to Hadamard product, $\mathbf{A}'$ is the adjacency matrix of the sampled sub-graph $\mathcal{G}' \sim \mathcal{T}_{\text{FeatRec}}(\mathcal{G})$, $\mathbf{M} \in \{0,1\}^{N' \times d}$ is the feature mask matrix whose rows equal to $\mathbf{1}$ if their corresponding nodes are targeted for reconstruction, and $\mathbf{W}^{\text{Dec}} \in \mathbb{R}^{d \times D}$ is the parameter matrix for the feature decoder. The objective for `FeatRec` is formulated as:

$$\mathcal{L}_{\text{FeatRec}} = \frac{||\hat{\mathbf{X}}' \odot \hat{\mathbf{M}} - \mathbf{X}' \odot \hat{\mathbf{M}}||_F}{||\mathbf{X}' \odot \hat{\mathbf{M}}||_F}, \tag{16}$$

where $\hat{\mathbf{M}} \in \{0,1\}^{N' \times D}$ is the mask matrix defined similarly to $\mathbf{M}$ with different dimension, and $\mathbf{X}' \in \mathbb{R}^{N' \times D}$ is the feature matrix of the sampled sub-graph $\mathcal{G}' \sim \mathcal{T}_{\text{FeatRec}}(\mathcal{G})$.

**Topological reconstruction**, denoted as `TopoRec`, aims at capturing the pair-wise relationships between the connected nodes. Given a sampled sub-graph $\mathcal{G}' \sim \mathcal{T}_{\text{TopoRec}}(\mathcal{G})$, we randomly select $B$ pairs of nodes $V^+ = \{(i,j)|\mathbf{A}'_{i,j} = 1\}$ and another $B$ pairs of nodes $V^- = \{(i,j)|\mathbf{A}'_{i,j} = 0\}$. The connection between two node $i$ and $j$ is measured by a logit calculated as:

$$P_{\text{TopoRec}}(i,j) = \sigma\big((f_g(\mathcal{G}'; \boldsymbol{\theta}_g)[i] \odot f_g(\mathcal{G}'; \boldsymbol{\theta}_g)[j]) \cdot \mathbf{W}^{\text{Topo}}\big), \tag{17}$$

where $[\cdot]$ refers to the indexing operation, and $\mathbf{W}^{\text{Topo}} \in \mathbb{R}^{d \times 1}$ is the parameter vector. The objective of `TopoRec` is maximizing the $P_{\text{TopoRec}}$ for nodes in $V^+$ and minimizing for nodes in $V^-$, formulated as a binary cross entropy loss as:

$$\mathcal{L}_{\text{TopoRec}} = -\frac{1}{2B} \sum_{(i,j) \in V^+} \log(P_{\text{TopoRec}}(i,j)) + \sum_{(i,j) \in V^-} \log(1 - P_{\text{TopoRec}}(i,j)). \tag{18}$$

## B.2 Whitening Decorrelation

**Representation decorrelation**, denoted as `RepDecor`, encourages the similarities between the representations of the same nodes in two independently augmented sub-graphs. During this process, to the prevent the node representations from collapsing into a trivial solution, the covariance between representations matrices of two sub-graphs are enforced to be an identity matrix, such that the knowledge learned by each dimension in the hidden space is orthogonal to each other (Ermolov et al., 2021; Zbontar et al., 2021; Zhang et al., 2021b). Given two sub-graphs $\mathcal{G}_1', \mathcal{G}_2' \sim \mathcal{T}_{\texttt{TopoRec}}(\mathcal{G})$ that are constituted by the same seed nodes but augmented differently, the objective of `RepDecor` is formulated as:

$$\mathcal{L}_{\texttt{RepDecor}} = \left\lVert f_g(\mathcal{G}_1'; \boldsymbol{\theta}_g) - f_g(\mathcal{G}_2'; \boldsymbol{\theta}_g) \right\rVert_F + \alpha \cdot \left\lVert f_g(\mathcal{G}_1'; \boldsymbol{\theta}_g)^\intercal \cdot f_g(\mathcal{G}_2'; \boldsymbol{\theta}_g)^\intercal - \mathbf{I}^{d \times d} \right\rVert_F, \quad (19)$$

where the first term encourages the node similarity and the second term regularize the solution from collapsing, $\mathbf{I}^{d \times d}$ is the square identity matrix with dimension $d \times d$, and $\alpha$ refers to a pre-defined balancing term (i.e., we use an $\alpha$ of 1e-3 across all datasets).

## B.3 Mutual Information Maximization

**Mutual information between nodes and the whole graph**, denoted as `MI-NG`, enables the graph encoder to learn coarse graph-level knowledge. Specifically, given a sampled sub-graph $\mathcal{G}' \sim \mathcal{T}_{\texttt{MI-NG}}(\mathcal{G})$, we first corrupt $\mathcal{G}'$ into $\mathcal{G}''$ by feature shuffling (Velickovic et al., 2019). Then we extract the hidden graph-level representation of $\mathcal{G}'$ by the graph mean pooling (Xu et al., 2018a), and enforce the representations of nodes in $\mathcal{G}'$ similar to the pooled representation while nodes in $\mathcal{G}''$ far away from the pooled representation. This pretext task allows the graph encoder to capture the perturbation brought by the topological change (i.e., feature shuffling). The objective of `MI-NG` is formulated as a binary cross entropy as:

$$\mathcal{L}_{\texttt{MI-NG}} = -\frac{1}{2N'} \sum_{i=1}^{N'} \log \left( P_{\texttt{MI-NG}}(f_g(\mathcal{G}'; \boldsymbol{\theta}_g)[i], \mathbf{h}_g) \right) + \log \left( 1 - P_{\texttt{MI-NG}}(f_g(\mathcal{G}''; \boldsymbol{\theta}_g)[i], \mathbf{h}_g) \right),$$

$$\text{s.t. } \mathbf{h}_g = \text{Pool}\left( f_g(\mathcal{G}'; \boldsymbol{\theta}_g) \right), \text{ and } P_{\texttt{MI-NG}}(\mathbf{h}, \mathbf{h}_g) = (\mathbf{h} || \mathbf{h}_g) \cdot \mathbf{W}^{\texttt{MI-NG}}, \quad (20)$$

where $\text{Pool}(\cdot)$ refers to the graph mean pooling function (Xu et al., 2018a), $||$ is the horizontal concatenation operation, and $\mathbf{W}^{\texttt{MI-NG}} \in \mathbb{R}^{2d \times 1}$ is the parameter vector.

**Mutual information between nodes and their sub-graphs**, denoted as `MI-NSG`, enables the graph encoder to learn fine-grained graph-level knowledge. Unlike `MI-NG` that enforces mutual information between node representations and the graph-level representation, `MI-NSG` maximizes the mutual information between the independently augmented sub-graphs entailed by the same anchor nodes, which learns fine-grained knowledge compared with `MI-NG`. Specifically, given two sub-graphs $\mathcal{G}_1', \mathcal{G}_2' \sim \mathcal{T}_{\texttt{TopoRec}}(\mathcal{G})$ that are constituted by the same seed nodes but augmented differently, the objective of `MI-NSG` is formulated as a variant of InfoNCE (Chen et al., 2020):

$$\mathcal{L}_{\texttt{MI-NSG}} = -\frac{1}{N'} \sum_{i=1}^{N'} \log \left( \frac{\text{Exp}\left( \text{Sim}\left( f_g(\mathcal{G}_1'; \boldsymbol{\theta}_g)[i], f_g(\mathcal{G}_2'; \boldsymbol{\theta}_g)[i] \right) / \tau \right)}{\sum_{j=1}^{N'} \text{Exp}\left( \text{Sim}\left( f_g(\mathcal{G}_1'; \boldsymbol{\theta}_g)[i], f_g(\mathcal{G}_1'; \boldsymbol{\theta}_g)[j] \right) / \tau \right) + \text{Exp}\left( \text{Sim}\left( f_g(\mathcal{G}_1'; \boldsymbol{\theta}_g)[i], f_g(\mathcal{G}_2'; \boldsymbol{\theta}_g)[j] \right) / \tau \right)} \right), \quad (21)$$

where $\text{Sim}(\mathbf{h}_1, \mathbf{h}_2) = \frac{\mathbf{h}_1 \cdot \mathbf{h}_2^\intercal}{||\mathbf{h}_1||_F \cdot ||\mathbf{h}_2||_F}$ is the similarity metric, $\text{Exp}(\cdot)$ stands for the exponential function, and $\tau$ is the temperature hyper-parameter used to control the sharpness of the similarity distribution (i.e., we explore a $\tau$ of 0.1 across all datasets).

## C Dataset Description

We evaluate our proposed PARETOGNN as well as unsupervised SSL-based GNNs on 11 real-world datasets spanning various fields such as citation network and merchandise network. Their statistics are shown in Table 4. For `Wiki-CS`, `Pubmed`, `Amazon-Photo`, `Amazon-Computer`, `Coauthor-CS`, and `Coauthor-Physics`, we use the API from Deep Graph Library (DGL)[1]

---

[1]`https://www.dgl.ai`

to load the datasets. For `ogbn-arxiv` and `ogbn-products`, we use the API from Open Graph Benchmark (OGB)[2]. For `Chameleon`, `Actor` and `Squirrel`, the datasets are downloaded from the official repository of Geom-GCN (Pei et al., 2019)[3].

Table 4: Dataset Statistics.

| Dataset | # Nodes | # Edges | # Features | Type | Split |
|---|---|---|---|---|---|
| Wiki-CS | 11,701 | 216,123 | 300 | Homophilous | 10%/10%/80% |
| Pubmed | 19,717 | 88,651 | 500 | Homophilous | 10%/10%/80% |
| Amazon-Photo | 7,650 | 119,043 | 745 | Homophilous | 10%/10%/80% |
| Amazon-Computer | 13,752 | 245,778 | 767 | Homophilous | 10%/10%/80% |
| Coauthor-CS | 18,333 | 81,894 | 6,805 | Homophilous | 10%/10%/80% |
| Coauthor-Physics | 34,493 | 247,962 | 8,415 | Homophilous | 10%/10%/80% |
| ogbn-arxiv | 169,343 | 1,166,243 | 128 | Homophilous | Public Split |
| ogbn-products | 2,449,029 | 61,859,140 | 100 | Homophilous | Public Split |
| Chameleon | 2,277 | 36,101 | 2,325 | Heterophilous | 10%/10%/80% |
| Actor | 7,600 | 33,544 | 931 | Heterophilous | 10%/10%/80% |
| Squirrel | 5,201 | 217,073 | 2,089 | Heterophilous | 10%/10%/80% |

# D    ADDITIONAL EXPERIMENTAL SETTINGS

**Hyper-parameters**. The hyper-parameters for PARETOGNN across all datasets are listed in Table 5.

Table 5: Hyper-parameters used for PARETOGNN. SAINT stands for sampling strategy proposed in GRAPHSAINT (Zeng et al., 2019), and we use its node version.

| Hyper-param. | Wiki.CS | Pubmed | Am.Photo | Am.Comp. | Co.CS | Co.Phy. | Cham. | Squirrel | Actor | arxiv | products |
|---|---|---|---|---|---|---|---|---|---|---|---|
| **Hyper-parameters w.r.t. `FeatRec`** | | | | | | | | | | | |
| Sampling Strategy | - | - | - | - | - | - | - | - | - | - | SAINT |
| # Seed Nodes | Full | Full | Full | Full | Full | Full | Full | Full | Full | Full | 200,000 |
| Node Mask Ratio | \multicolumn{11}{c}{0.5 used for all datasets} | | | | | | | | | | |
| Edge Drop Ratio | \multicolumn{11}{c}{0.35 used for all datasets} | | | | | | | | | | |
| **Hyper-parameters w.r.t. `TopoRec`** | | | | | | | | | | | |
| Sampling Strategy | \multicolumn{11}{c}{$K$-order sage sampler (Hamilton et al., 2017) ($K$ equals to the number of convolution layers in the GNN encoder)} | | | | | | | | | | |
| Batch size $B$ | 10,240 | 10,240 | 5,120 | 10,240 | 10,240 | 10,240 | 10,240 | 10,240 | 10,240 | 5,120 | 5,120 |
| **Hyper-parameters w.r.t. `RepDecor`** | | | | | | | | | | | |
| Sampling Strategy | \multicolumn{11}{c}{SAINT used for all datasets} | | | | | | | | | | |
| # Seed Nodes | 10,000 | 10,000 | 5,000 | 10,000 | 10,000 | 20,000 | 1,500 | 5,000 | 5,000 | 20,000 | 100,000 |
| Edge Drop Ratio | 0.2 | 0.2 | 0.2 | 0.2 | 0.5 | 0.5 | 0.2 | 0.2 | 0.2 | 0.2 | 0.2 |
| Feature Mask Ratio | 0.2 | 0.2 | 0.2 | 0.2 | 0.5 | 0.5 | 0.2 | 0.2 | 0.2 | 0.2 | 0.2 |
| **Hyper-parameters w.r.t. `MI-NG`** | | | | | | | | | | | |
| Sampling Strategy | \multicolumn{11}{c}{$k$-order sub-graphs} | | | | | | | | | | SAINT |
| $k$ | Full | Full | 3 | Full | 3 | 2 | 3 | 3 | 3 | 3 | - |
| # Seed Nodes | Full | Full | 5,120 | Full | 10,240 | 10,240 | 1,024 | 3,072 | 5,120 | 10,240 | 20,480 |
| **Hyper-parameters w.r.t. `MI-NSG`** | | | | | | | | | | | |
| Sampling Strategy | \multicolumn{11}{c}{$k$-order sub-graphs} | | | | | | | | | | SAINT |
| $k$ | 3 | 3 | 3 | 3 | 3 | 1 | 3 | 3 | 3 | 1 | - |
| # Seed Nodes | 5,120 | 5,120 | 5,120 | 5,120 | 10,240 | 5,120 | 1,024 | 3,072 | 3,072 | 512 | 20,480 |
| Edge Drop Ratio | 0.2 | 0.2 | 0.2 | 0.2 | 0.5 | 0.5 | 0.2 | 0.2 | 0.2 | 0.2 | 0.2 |
| Feature Mask Ratio | 0.2 | 0.2 | 0.2 | 0.2 | 0.5 | 0.5 | 0.2 | 0.2 | 0.2 | 0.2 | 0.2 |
| **Hyper-parameters w.r.t. the GNN encoder** | | | | | | | | | | | |
| # Layers | 2 | 2 | 2 | 2 | 2 | 2 | 2 | 2 | 2 | 3 | 3 |
| Hidden Dimension | [512, 256] | [512, 256] | [512, 256] | [512, 256] | [512, 256] | [512, 256] | [512, 256] | [512, 256] | [512, 256] | 512×3 | 256×3 |
| Activation | \multicolumn{11}{c}{PReLU used for all datasets} | | | | | | | | | | |
| Batch Norm. | Y | Y | Y | Y | Y | Y | Y | Y | Y | N | N |
| Layer Norm. | N | N | N | N | N | N | N | N | N | Y | Y |
| Optimizer | \multicolumn{11}{c}{AdamW with 1e-5 weight decay used for all datasets} | | | | | | | | | | |
| Training Steps | \multicolumn{11}{c}{10,000 used for all datasets} | | | | | | | | | | |
| $\xi$ Stopping constant | \multicolumn{11}{c}{1e-5 used for all datasets} | | | | | | | | | | |
| Learning Rate | 5e-4 | 1e-3 | 1e-4 | 5e-4 | 5e-5 | 1e-4 | 5e-5 | 1e-3 | 1e-3 | 1e-4 | 1e-4 |

**Hardware and software configurations**. We conduct experiments on a server having one RTX3090 GPU with 24 GB VRAM. The CPU we have on the server is an AMD Ryzen 3990X with 128GB RAM. The software we use includes DGL 1.9.0 and PyTorch 1.11.0.

---

[2] https://ogb.stanford.edu
[3] https://github.com/graphdml-uiuc-jlu/geom-gcn

**Baseline Implementation**. As for the baseline models that we compare PARETOGNN with, we explore the implementations provided by code repositories listed as follows:

- DGI (Velickovic et al., 2019): `https://github.com/dmlc/dgl/tree/master/examples/pytorch/dgi`.
- GRACE (Zhu et al., 2020): `https://github.com/dmlc/dgl/tree/master/examples/pytorch/grace`.
- MVGRL (Hassani & Khasahmadi, 2020): `https://github.com/dmlc/dgl/tree/master/examples/pytorch/mvgrl`.
- AUTOSSL (Jin et al., 2022): `https://github.com/ChandlerBang/AutoSSL`.
- BGRL (Thakoor et al., 2022): `https://github.com/dmlc/dgl/tree/master/examples/pytorch/bgrl`.
- CCA-SSG (Zhang et al., 2021b): `https://github.com/hengruizhang98/CCA-SSG`.
- GRAPHMAE (Hou et al., 2022): `https://github.com/THUDM/GraphMAE`.

We sincerely appreciate the authors of these works for open-sourcing their valuable code and researchers at DGL for providing reliable implementations of these models.

## E  TRAINING TIME AND MEMORY CONSUMPTION

The scalability w.r.t. the graph dimensions is well leveraged by our utilization of sampled sub-graphs and experimentally verified by PARETOGNN's strong performance over large graphs, as shown in Table 3. On top of this, we also measure the training time and GPU memory consumption to give a direct empirical understanding of PARETOGNN's overhead, as shown in Table 6. For AUTOSSL, we notice that the calculation of the pseudo-homophily is extremely slow because such a process cannot enjoy the GPU acceleration. GPU remains mostly idle during the training of AUTOSSL. Though PARETOGNN is not as efficient as BGRL when the graphs are small-scaled; for large graphs such as OGBN-PRODUCTS, all methods require sampling strategies and in this case the efficiency of PARETOGNN is on par with that of BGRL. BGRL learns from one large graph (though sampled), and PARETOGNN learns from multiple relatively small graphs, which entails similar computational overheads.

Table 6: Training time and memory consumption of PARETOGNN. *: model is trained on sub-graphs with dimensions matching the maximum GPU memory (i.e., 24 GB).

| Dataset | AUTOSSL | GRACE | BGRL | PARETOGNN |
|---|---|---|---|---|
| Training time per 1,000 iterations (s) | | | | |
| `Wiki-CS` | 7,319 | 197 | 34 | 234 |
| `Co.CS` | 11,340 | 662 | 121 | 618 |
| `ogbn-arxiv` | $1.76 \times 10^5$ | 844* | 331 | 690 |
| `ogbn-products` | OOM | 1,573* | 1,108* | 1,134 |
| Peak Memory Consumption (GB) | | | | |
| `Wiki-CS` | 15.7 | 6.3 | 1.9 | 12.1 |
| `Co.CS` | 23.7 | 13.8 | 5.4 | 12.6 |
| `ogbn-arxiv` | 23.7 | 23.8* | 8.9 | 21.4 |
| `ogbn-products` | OOM | 24.0* | 23.7* | 22.8 |

## F  PERFORMANCE OF INDIVIDUAL TASKS ON LARGE GRAPHS

In Table 3, we demonstrate the task generalization of all models over large graphs (i.e., `ogbn-arxiv` and `ogbn-products`) quantified by the average performance over the four downstream tasks. Here

Table 7: The performance and task generalization of PARETOGNN as well as state-of-the-art unsupervised baselines over large graphs. (*: Graphs are sampled by GRAPHSAINT (Zeng et al., 2019) matching the memory of others due to OOM.)

| Dataset | Method | Node Clas. | Node Clus. | Link Pred. | Part. Pred. | Average |
|---------|--------|-----------|-----------|-----------|------------|---------|
| ogbn-arxiv | DGI | $70.26_{\pm 0.22}$ | $42.46_{\pm 0.17}$ | $\mathbf{98.02}_{\pm 0.04}$ | $68.27_{\pm 0.20}$ | 69.75 |
| | GRACE * | $71.04_{\pm 0.23}$ | $42.01_{\pm 0.12}$ | $95.12_{\pm 0.09}$ | $67.11_{\pm 0.43}$ | 68.82 |
| | AUTOSSL | $69.13_{\pm 0.04}$ | $41.70_{\pm 0.00}$ | $96.12_{\pm 0.01}$ | $65.12_{\pm 0.41}$ | 68.02 |
| | BGRL | $\mathbf{71.55}_{\pm 0.04}$ | $44.25_{\pm 0.10}$ | $95.05_{\pm 0.03}$ | $69.32_{\pm 0.11}$ | 70.04 |
| | CCA-SSG | $71.24_{\pm 0.07}$ | $\underline{42.45}_{\pm 0.07}$ | $96.63_{\pm 0.02}$ | $68.17_{\pm 0.12}$ | 69.62 |
| | GRAPHMAE | $71.21_{\pm 0.13}$ | $44.21_{\pm 0.12}$ | $96.02_{\pm 0.17}$ | $\mathbf{73.14}_{\pm 0.23}$ | $\underline{71.15}$ |
| | PARETOGNN | $71.47_{\pm 0.09}$ | $\mathbf{46.71}_{\pm 0.11}$ | $\underline{97.66}_{\pm 0.02}$ | $69.70_{\pm 0.16}$ | $\mathbf{71.39}$ |
| ogbn-products | DGI * | $72.52_{\pm 0.21}$ | $50.00_{\pm 0.20}$ | $98.81_{\pm 0.13}$ | $80.59_{\pm 0.12}$ | $\underline{75.48}$ |
| | GRACE * | $\underline{72.65}_{\pm 0.14}$ | $\underline{50.12}_{\pm 0.22}$ | $97.96_{\pm 0.14}$ | $81.59_{\pm 0.23}$ | $\underline{75.58}$ |
| | AUTOSSL | | | OOM | | |
| | BGRL * | $72.11_{\pm 0.24}$ | $49.87_{\pm 0.12}$ | $\mathbf{98.89}_{\pm 0.09}$ | $79.87_{\pm 0.26}$ | 75.19 |
| | CCA-SSG * | $72.09_{\pm 0.24}$ | $47.78_{\pm 0.31}$ | $94.28_{\pm 0.11}$ | $77.50_{\pm 0.10}$ | 72.92 |
| | GRAPHMAE * | $72.60_{\pm 0.14}$ | $47.08_{\pm 0.07}$ | $\underline{98.87}_{\pm 0.10}$ | $80.99_{\pm 0.05}$ | 74.89 |
| | PARETOGNN * | $\mathbf{73.25}_{\pm 0.11}$ | $\mathbf{50.17}_{\pm 0.32}$ | $98.54_{\pm 0.13}$ | $\mathbf{81.97}_{\pm 0.23}$ | $\mathbf{75.98}$ |

we provide additional experimental results on models' performance of every individual task, as shown in Table 7.

We notice that the graph dimension is not a limiting factor for the strong task generalization of our proposal. Besides the conclusion we have drawn in Section 3.4, where PARETOGNN outperforms the runner-ups by 2.5 on rank of the average performance calculated over the four tasks, we also observe strong single-task performance on some tasks, demonstrating that PARETOGNN achieves better task generalization via the disjoint yet complementary knowledge learned from different philosophies.

## G  SADDLE-POINT TEST FOR PARAMETERS IN PARETOGNN

In PARETOGNN, the saddle-point test conditions (Désidéri, 2012) for the shared GNN encoder (i.e., $f_g(\cdot; \boldsymbol{\theta}_g)$) as well as the task-specific heads (i.e., $f_k(\cdot; \boldsymbol{\theta}_k)$) are defined as the following:

- For $\boldsymbol{\theta}_g$, there exist $\boldsymbol{\alpha}$ such that every element in $\boldsymbol{\alpha}$ is greater than or equal to 0, $||\boldsymbol{\alpha}|| = 1$, and $\sum_{k=1}^{K} \alpha_k \cdot \nabla_{\boldsymbol{\theta}_g} \mathcal{L}_k(\mathcal{G}; \mathcal{T}_k, \boldsymbol{\theta}_g, \boldsymbol{\theta}_k) = 0$.
- For $\{\boldsymbol{\theta}_k\}_{k=1}^{K}$, we have $\nabla_{\boldsymbol{\theta}_k} \mathcal{L}_k(\mathcal{G}; \mathcal{T}_k, \boldsymbol{\theta}_g, \boldsymbol{\theta}_k) = 0$.

According to MGDA, the solution above gives a descent direction that improves all tasks, which improves the task generalization while minimizing potential conflicts.

## H  CONNECTIONS TO OTHER PARETO LEARNING FRAMEWORKS

This section is greatly inspired by valuable comments from the reviewers of this paper[4]. We sincerely appreciate endeavors from the reviewers to help us further refining this paper.

Task reconciliation by Pareto learning is mostly explored by works in the Computer Vision community (Lin et al., 2019; Mahapatra & Rajan, 2020; Chen et al., 2021). In their settings, usually there exist one main task and multiple auxiliary tasks. They explore Pareto learning with preference vectors to enforce the learning models to remain good performance on the main task while extracting information from auxiliary tasks as much as possible. Pareto learning with user-defined preference vectors is sensible in their cases, as they do not want the auxiliary tasks to hinder the learning of the main task (i.e., a preference of the main task over the auxiliary tasks). For example, PSST (Chen et al., 2021) enhances the model's few-shot learning capability (i.e., the main task) by optimizing the model with the main task as well as additional pretext tasks (i.e., the auxiliary tasks). In this case,

---

[4]Detailed reviews are available at: https://openreview.net/forum?id=1tHAZRqftM.

while remaining competitive performance on the main task, PSST also regularizes the learning model against extracting task-irrelevant information (Ren & Lee, 2018; Ravanelli et al., 2020) by training with auxiliary tasks, which empirically further improves the performance on the main task.

However, our scenario is completely different from theirs because we do not want a task governing others (i.e., all tasks are main tasks). The task reconciliation proposed in our paper is not biased toward any task. If preference vectors are explored, some pretext tasks will definitely be jeopardized due to the definition of Pareto optimality (i.e., Definition 1). Such a bias over the pretext tasks would cause the self-supervised GNNs not performing equally well on every downstream task and dataset. It hurts the overall average performance and task generalization, which are the main focuses of this paper. To empirically validate that an utilization of a preference vector contradicts our goal, we conduct experiments on PARETOMTL (Lin et al., 2019) with five preference vectors (i.e., five vectors for different preferences over five tasks), as shown in Table 8.

Table 8: The task generalization of multi-task learning with weighted summation (i.e., `w/o Pareto`), PARETOGNN, and five PARETOMTL variants with preferences favoring different tasks.

| Dataset | w/o Pareto | PARETOGNN | PARETOMTL with FeatRec | PARETOMTL with TopoRec | PARETOMTL with RepDecor | PARETOMTL with MI-NG | PARETOMTL with MI-NSG |
|---|---|---|---|---|---|---|---|
| WIKI.CS | 74.64 | **76.03** | 74.15 | 72.42 | 70.36 | 72.44 | 75.89 |
| PUBMED | 69.82 | **72.48** | 70.66 | 67.92 | 67.59 | 68.30 | 69.01 |

We observe that multi-task self-supervised learning via Pareto learning with a preference vector under most of the time does not even outperform the vanilla weighted summation (i.e., `w/o Pareto`) due to the bias introduced by the preference vectors. Our proposed PARETOGNN consistently outperforms variants based on PARETOMTL with different preferences. Though sometimes variants with preference vectors approach the performance of PARETOGNN (i.e., PARETOMTL with `MI-NSG` in WIKI.CS), coming to this results needs a grid-search on all possible preference vectors, which is inefficient when heuristics such as the prior knowledge are not available. Including preferences in multi-task learning could be very helpful when one or a set of training tasks need to be highlighted, whose success is demonstrated by the experiments in PARETOMTL. But this is not our case, because to achieve strong task generalization for GNNs, philosophies contained in all pretext tasks are equally important. Our implementation of PARETOMTL comes from its official Github repository[5].

---

[5] https://github.com/Xi-L/ParetoMTL

