# OpenReview forum: "Multi-task Self-supervised Graph Neural Networks Enable Stronger Task Generalization"
_ICLR.cc/2023/Conference — ICLR 2023 poster_

### Official Review · Reviewer_15ww · 2022-10-24

**Confidence:** 3
**Correctness:** 4
**Technical Novelty And Significance:** 3
**Empirical Novelty And Significance:** 3
**Recommendation:** 6

**Clarity, Quality, Novelty And Reproducibility:**

The paper is clearly written and to me seems to tick off the quality and novelty requirements. Code is also provided.

**Strength And Weaknesses:**

(+) The question represents an interesting topic to be investigated and the paper presents a reasonable approach which, to the best of my knowledge, is novel, especially the integration of the gradients.

(+) The authors provide theory proving convergence. Please note that I did not check all that in full detail.

(+) The evaluation considers various datasets, also challenging ones from OGB, tasks, and baselines. I am no expert in SSL to be able to judge if the most important competitors are considered.

(-) Some statements seem unlucky/misleading to me:
- The main question "Does combining multiple philosophies enhance task generalization for SSL-based GNNs?" sounds a bit too easy. "Yes" should be a very likely I would think.
- Similarly, the authors claim throughout the paper that models focusing on a single task perform "poorly" (as narrow experts) and generalization beyond a single task is needed. I would think that these are too bold claims. If we developed perfect experts over time, this would not be a problem. Nevertheless, the fact that the paper shows that the generalization obtained helps improving over single-task-focused models is great. I just would frame it differently.

(-) The paper is missing related work. For example, [1] propose to combine different self-supervised learning strategies and demonstrate that the combination is beneficial. Similarly, I would consider multi-task learning in GNNs as related work to be mentioned in more detail.
p.9 "In these frameworks" - if there are more, these should be referenced as well since this is closely related work.

[1] Hu et al. STRATEGIES FOR PRE-TRAINING GRAPH NEURAL NETWORKS, ICLR 2020



**Summary Of The Paper:**

The paper investigates the use of multi-task learning for GNN self-supervised learning and particularly uses a combined gradient based on pareto optimality. The model seems to improve upon existing ones, interestingly, especially also in terms of generalization capability w.r.t. heterophily/homophily.


**Summary Of The Review:**

Altogether, this work seems to be a solid contribution to me. I am too few expert of self-supervised learning to judge if it is really novel w.r.t. SOTA. If the other reviewers can confirm the latter, I would suggest acceptance.

---

> ### Author Response · Authors · 2022-11-12
> **Response to Reviewer 15ww [2/2]**
>
> ### **Missing related works**
> > Thank you for pointing out this missing related work. We have included and discussed this work in our updated manuscript (i.e., in the location that you recommended).
>
> ### **State-of-the-art performance**
> > The comprehensiveness of our experiments is unanimously acknowledged by all reviewers. We believe that the task generalization of ParetoGNN is strong since most of our baselines come from very recent top-tier venues (e.g., CCA-SSG [1] from NeurIPS'21, BGRL [2] and AutoSSL [3] from ICLR'22, and GraphMAE [4] from KDD'22, which was released only two months ahead of ICLR's deadline). ParetoGNN achieves the best task generalization over four tasks and eleven datasets by comparing these recent and high-quality works.
>
> We modified the manuscript to clarify your concerns, and we hope we have satisfactorily answered your questions. If so, could you please consider increasing your rating? If you have remaining doubts and concerns, please let us know, and we will happily respond. Thank you!
>
> Best regards,\
> ParetoGNN authors
> ***
> [1] Zhang et al., From Canonical Correlation Analysis to Self-supervised Graph Neural Networks. NeurIPS 2021.\
> [2] Thakoor et al., Large-scale Representation Learning on Graphs via Bootstrapping. ICLR 2022.\
> [3] Jin et al., Automated Self-supervised Learning for Graphs. ICLR 2022.\
> [4] Hou et al., GraphMAE: Self-Supervised Masked Graph Autoencoders. KDD 2022.

---

> ### Author Response · Authors · 2022-11-12
> **Response to Reviewer 15ww [1/2]**
>
> Dear Reviewer 15ww:\
> Thank you for your valuable feedback. We sincerely appreciate your acknowledgment of the motivations behind our model design, the theoretical analysis, and the insights and comprehensiveness of our experiments.
> Our detailed response to your concern about the novelty is listed as follows:
> ### **Explanation of the two concerning statements**
> > **Statement 1**: _Does combining multiple philosophies enhance the task generalization for SSL-based GNNs?_
> >
> > Though its rationale is intuitively simple, coming up with the answer to this question is non-trivial and requires comprehensive empirical studies upon existing SSL-based GNNs over many downstream tasks and datasets. Actually, as shown in Table 1, simply combining multiple philosophies by weighted summation (denoted as w/o Pareto) **does not always enhances the task generalization**.
> >
> > Inspired by your comments, we think changing this question to “How to combine multiple philosophies to enhance the task generalization for SSL-based GNNs” would arouse more thinking from our readers.
> >
> > According to our comprehensive empirical evaluations, existing SSL-based GNNs perform good on only one task or two and their good performance usually does not translate into strong task generalization across various downstream tasks or datasets (e.g., DGI, based on mutual information maximization, excels at partition prediction but underperforms on node classification and link prediction). For now there does not exist any work that formally studies this problem and correspondingly bridges the gap (i.e., sub-optimal task generalization). Since our proposed ParetoGNN enhances the task generalization by combining multiple philosophies via Pareto learning, we ask this question early in the introduction section to further emphasize our motivation.
> >
> > **Statement 2**: _poor task generalization of single-task models._
> >
> > Thank you for your suggestions, which helped us rephrase and clarify some conclusions we made. In the Conclusion section, we stated "performs poorly in this setting", which we agree is a bit too bold. We rephrased it into a more accurate description: "their promising performance on one task or two usually does not translate into good task generalization across various downstream tasks and datasets".
> >
> > When focusing on a single downstream task or a specific dataset, a narrow expert that explores only a single philosophy might not be problematic. It is possible that some downstream tasks are strongly correlated to one pretext task but not others (i.e., pretext tasks based on feature reconstruction works well on the node classification task when the node features are informative such as Coauthor-CS and Coauthor-Physics). However, as the intent behind SSL is generalization across tasks, our paper tackles this by showing that existing SSL methods may not generalize well across multiple downstream tasks and datasets and accordingly proposing solutions for this setting. Indeed, as you have mentioned, in some cases, having a "perfect expert" is just fine.
> >
> > Besides, designing a ''perfect SSL task'' spanning various philosophies is difficult. It is possible that people would come up with new downstream tasks that the perfect SSL task does not suffice. When such a new downstream task comes, designing a ''more perfect task'' would be very expensive. However, in our case, we can straightforwardly include a list of simple pretext tasks that covers all required philosophies. As we have noted in the paper, ParetoGNN is not limited to the current learning objectives and the incorporation of other philosophies is a straightforward extension.

---

> ### Comment · Reviewer_15ww · 2022-11-16
> **Thank you for the detailed and clarifying response.**
>
> I definitely will consider to increase my rating after the discussion phase.

---

> > ### Author Response · Authors · 2022-11-17
> > **Thank you!**
> >
> > We sincerely appreciate your willingness to consider raising the recommendation and we are happy to discuss if you have any further questions or concerns.

---

### Official Review · Reviewer_WfQN · 2022-10-24

**Confidence:** 4
**Correctness:** 4
**Technical Novelty And Significance:** 3
**Empirical Novelty And Significance:** 4
**Recommendation:** 6

**Clarity, Quality, Novelty And Reproducibility:**

The clarity, quality, and reproducibility are good, while the novelty is somewhat limited.

**Strength And Weaknesses:**

Pros:
[+] Multi-task SSL for node representation learning over graphs is a novel and important problem.
[+] This paper shows that task conflicts commonly exist in real-world graph datasets and theoretically proves that the proposed method can resolve task conflicts.
[+] This paper conducts extensive experiments over four downstream tasks and 11 benchmark datasets.
[+] The illustration of motivation, method design, and experiments are quite clear.
[+] Experiment details are contained in the appendix, including hyperparameters, settings, etc. Source codes are also included.

Cons:
[-] It seems that the proposed method to resolve task conflicts is agnostic of domains, i.e., not specifically designed for graphs. It would make the paper stronger if further analysis of the difference between conflicts in graphs and other domains could be analyzed. For this reason, I think that the technical novelty is somewhat limited.

=== after rebuttal ===
I have read the rebuttal and thank the authors for further clarifications. All things considered (including the quality of my other reviewed papers), I'd like to keep a slight positive score as 6.


**Summary Of The Paper:**

This paper proposes a multi-task self-supervised learning framework namely ParetoGNN for node representation learning on graphs. The proposed method adopts five pretext self-supervised learning tasks and designs a multiple-gradient descent algorithm to reconcile different tasks by actively learning and minimizing conflicts. Extensive experiments over four downstream tasks and 11 benchmark datasets show that the proposed method can improve the model's task generalization ability.

**Summary Of The Review:**

See above.

---

> ### Author Response · Authors · 2022-11-12
> **Response to Reviewer WfQN**
>
> Dear Reviewer WfQN:\
> Thank you for your valuable feedback. We sincerely appreciate your acknowledgment of the writing quality, the motivations behind our model design, and the insights and comprehensiveness of our experiments.
> Our detailed response to your concerns is listed as follows:
> ### **Clarification on the novelty of ParetoGNN**
> > We appreciate your pointing out this confusion, and add a section in the appendix (Appendix H in page 19) which includes relevant points from this response.
> >
> > **[Pareto learning in other domains]** Task reconciliation by Pareto learning is mostly explored by works in the Computer Vision community [1,2,3]. In their settings, usually there exist one main task and multiple auxiliary tasks. They explore Pareto learning with preference vectors to enforce the learning models to remain good performance on the main task while extracting information from auxiliary tasks as much as possible. Pareto learning with user-defined preference vectors is sensible in their cases, as they do not want the auxiliary tasks to hinder the learning of the main task (i.e., a preference of the main task over the auxiliary tasks).
> >
> > **[Pareto learning in our setting]** However, our scenario is completely different from theirs because we do not want a task governing others (i.e., all tasks are main tasks). The task reconciliation proposed in our paper is not biased toward any task. If preference vectors are explored, some pretext tasks will definitely be jeopardized due to the definition of Pareto optimality. Such a bias over the pretext tasks would cause the self-supervised GNNs not performing equally well on every downstream task and dataset. It hurts the overall average performance and task generalization, which are the main focuses of this paper. We conduct additional experiments on Pareto learning with preference vectors (i.e., Table 8 in our revision), and empirically show that an utilization of preference vectors is not appropriate in our setting. Besides, we also theoretically demonstrate the convergence guarantee of our Pareto learning algorithm, which to the best of our knowledge is not discussed in any of the Pareto papers.
> >
> >**[Theoretical convergence guarantee]**: We also provide a complete theorem (i.e., Theorem 1) upon the convergence guarantee of our iterative search algorithm, which to the best of our knowledge is not discussed in any of the Pareto papers. We theoretically proved that the convergence rate of our algorithm is negatively proportional to the number of iterations (i.e., $O(1/\gamma)$), and the error of the induced task reconciliation between all tasks is at most $4\beta/(\gamma+1)$. According to this theorem and our empirical observation, ParetoGNN finds the Pareto optimal solution with less than 100 iterations, which is fast and affordable.
> >
> > **[Novelty besides Pareto Learning]** Beside the novelties with respect to the Pareto learning mentioned above, an important contribution of our work is to analyze the task generalization of the existing SSL-based GNNs in a more rigorous setting. By conducting experiments over four downstream tasks and eleven benchmarks, we empirically demonstrated that the promising performance of existing SSL-based GNNs on one task or two usually does not translate into good task generalization. This observation could bring people's attention to the problems underlying the current trending evaluations of SSL-based GNNs and further shed light on potential solutions (like the multi-task self-supervised learning we explored in this work).
> >
> > **[Strong Empirical Contribution]**: Self-supervised by multiple tasks spanning various philosophies via Pareto learning, ParetoGNN achieves the **best task generalization** over four trending graph downstream tasks and eleven community acknowledged benchmark datasets, by comparing seven state-of-the-art SSL-based GNNs (i.e., top ranked average performance on all datasets with up to 5.33 improvement as shown in Table 2). We believe the good performance of ParetoGNN and insightful empirical observations stand on its own to make great contributions to the graph learning community.
>
> We modified the manuscript to clarify your concerns, and we hope we have satisfactorily answered your questions. If so, could you please consider increasing your rating? If you have remaining doubts and concerns, please let us know, and we will happily respond. Thank you!
>
> Best regards,\
> ParetoGNN authors
> ***
> [1] Chen et al., Pareto Self-Supervised Training for Few-Shot Learning. CVPR 2021.\
> [2] Lin et al., Pareto Multi-Task Learning. NeurIPS 2019.\
> [3] Mahapatra et al., Multi-task Learning with User Preferences: Gradient Descent with Controlled Ascent in Pareto Optimization. ICML 2020.

---

### Official Review · Reviewer_wz1H · 2022-10-25

**Confidence:** 2
**Clarity, Quality, Novelty And Reproducibility:** I asked novelty issue on the above se…
**Correctness:** 3
**Technical Novelty And Significance:** 2
**Empirical Novelty And Significance:** 2
**Recommendation:** 6

**Strength And Weaknesses:**

**Strengths**


- The paper including the appendix is well-written and organized.

- The authors compare several features of the existing methods with the proposed method and provide a detailed discussion.

- The results on benchmarks are comparable to or better than most baselines.

---


There has been papers that explore Pareto optimality for SSL or MTL [1,2].

What is the main contribution of your work compared to them?

Of course I could have noticed, but I'm asking for the benefit of other readers.

Because it might seem to just simple extension of these series of works into graph version.

In Pareto MTL [2], the authors claim that the multiple gradient descent algorithm (MGDA) is not capable to incorporate different trade-off preference information as illustrated in Figure 2 of [2].

Does it mean ParetoGNN also can have the same weakness?

---


[1] Pareto Self-Supervised Training for Few-Shot Learning, Chen et al., CVPR 2021

[2] Pareto Multi-Task Learning, Lin et al., NeurIPS 2019

**Summary Of The Paper:**

The authors propose PARETOGNN, which is a multi-task SSL framework using the concept of Pareto optimality.


The self-supervised learning (SSL) for multi-task is still challenging especially over graph-structured data.


In this environment, the SSL frameworks for the graph that adhere to just one objective might perform poorly to the others.


To improve task generalization, PARETOGNN is simultaneously trained on various SSL tasks while promoting Pareto optimality using the Multiple Gradient Descent Algorithm(MGDA).


It allows the model to capture inherent patterns that are relevant to several pretext tasks while reducing possible conflicts.


As a result, the model learns different patterns transferable to multiple tasks on graphs.

**Summary Of The Review:**

I set the confidence of this review to a low score because I would change (or finalize) my decision after the discussion period (regarding the novelty issue).

---

> ### Author Response · Authors · 2022-11-12
> **Response to Reviewer wz1H [2/2]**
>
> ### **Question: What is the main contribution of ParetoGNN compared with PSST [1] and ParetoMTL [2]?**
> >
> > ParetoMTL [2] and PSST [1] also explore the concept of Pareto optimality to reconcile different tasks. The novelties of the proposed ParetoGNN lie in three perspectives (i.e., the *technical contribution*, the *theoretical convergence guarantee*, and the *motivation* of promoting Pareto optimality).
> > * **[Technical contribution]**: PSST and ParetoMTL achieve the Pareto optimal solution with respect to user-defined preference vectors, which regularize the model to have a preference over one or a set of training tasks; whereas ParetoGNN makes no prior assumption on the task preferences. As demonstrated in Figure 1, to promote Pareto optimality and further enhance task generalization, ParetoGNN reconciles different pretext tasks by finding the minimum-norm vector in the entire convex hull, which gives a Pareto optimal solution not biased toward any pretext task. On the other hand, for ParetoMTL or PSST with user-defined preference vectors, the minimum-norm vector is found in a submanifold of the convex hull, which gives a Pareto optimal solution biased toward a set of pretext tasks. It hurts the task generalization in our setting since all pretext tasks are equally important, which is empirically proved in our experiment section and Table Re1.
> >
> > * **[Theoretical convergence guarantee]**: We also provide a complete theorem (i.e., Theorem 1) upon the convergence guarantee of our iterative search algorithm, which to the best of our knowledge is not discussed in any of the Pareto papers. We theoretically proved that the convergence rate of our algorithm is negatively proportional to the number of iterations (i.e., $O(1/\gamma)$), and the error of the induced task reconciliation between all tasks is at most $4\beta/(\gamma+1)$. According to this theorem and our empirical observation, ParetoGNN finds the Pareto optimal solution with less than 100 iterations, which is fast and affordable.
> >
> > * **[Difference in the motivation and application setting]**: As mentioned in our response to the first question, in the setting of ParetoMTL and PSST, Pareto learning with user-defined preference vectors is sensible, as they do not want the auxiliary tasks to hinder the learning of the main task (i.e., a preference of the main task over the auxiliary tasks). However, this setting is not applicable to our scenario, because we do not want a task governing others (i.e., all tasks are main tasks). Each pretext task reflects a philosophy that benefits a set of downstream tasks or datasets. Hence an adaptation of a preference in our multi-task self-supervised learning hurts the performance on downstream tasks and datasets that the jeopardized pretext tasks favor. Inspired by the concept of Pareto optimality, we propose a multi-task self-supervised learning framework for graph neural networks to enhance the task generalization over multiple downstream tasks and datasets, by promoting the Pareto optimality that is not biased toward any pretext tasks.
> >
> > * **[Strong Empirical Contribution]**: Self-supervised by multiple tasks spanning various philosophies via Pareto learning, ParetoGNN achieves the **best task generalization** over four trending graph downstream tasks and eleven community acknowledged benchmark datasets, by comparing seven state-of-the-art SSL-based GNNs (i.e., top ranked average performance on all datasets with up to 5.33 improvement). We believe the good performance of ParetoGNN and insightful empirical observations stand on their own to make great contributions to the graph learning community.
>
> ### **More clarification on the novelty of ParetoGNN**
> > **[Novelty besides Pareto Learning]** Beside the three novelties with respect to the Pareto learning mentioned above, another important contribution of our work is to analyze the task generalization of the existing SSL-based GNNs in a more rigorous setting. By conducting comprehensive experiments over four downstream tasks and eleven benchmarks, we empirically demonstrated that the promising performance of existing SSL-based GNNs on one task or two usually does not translate into good task generalization. This observation could bring people's attention to the problems underlying the current trending evaluations of SSL-based GNNs and further shed light on potential solutions (like the multi-task self-supervised learning we explored in this work).
>
> We modified the manuscript to clarify your concerns, and we hope we have satisfactorily answered your questions. If so, could you please consider increasing your rating? If you have remaining doubts and concerns, please let us know, and we will happily respond. Thank you!
>
> Best regards,\
> ParetoGNN authors
> ***
> [1] Chen et al., Pareto Self-Supervised Training for Few-Shot Learning. CVPR 2021.\
> [2] Lin et al., Pareto Multi-Task Learning. NeurIPS 2019.\
> [3] https://github.com/Xi-L/ParetoMTL

---

> ### Author Response · Authors · 2022-11-12
> **Response to Reviewer wz1H [1/2]**
>
> Dear Reviewer wz1H:\
> Thank you for your valuable feedback. We sincerely appreciate your acknowledgment of the writing quality, insights from our experiments, and performance improvement.
> Our detailed response to your concerns is listed as follows:
> ### **Question: Does ParetoGNN have the same weakness (i.e., not capable of incorporating different trade-off preference vectors)?**
> > We appreciate your pointing out this confusion, and add a section in the appendix (Appendix H in page 19) which includes relevant points from this response.
> >
> > **[Why not a weakness for ParetoGNN]** The weakness you refer to is that ParetoGNN cannot incorporate pre-defined trade-offs between pretext tasks.
> The two papers that you pointed out [1,2] are designed for settings where there exist a main task and multiple auxiliary tasks.
> In this setting, Pareto learning with user-defined preference vectors is sensible, as they do not want the auxiliary tasks to hinder the learning of the main task (i.e., a preference of the main task over the auxiliary tasks). However, our scenario is completely different from theirs because we do not want a task governing others (i.e., all tasks are main tasks).
> >
> > ParetoGNN indeed does not accept user-defined preference vectors and this is **not a weakness or design flaw in our setting**.
> In fact, a preference vector over multiple pretext tasks actually **contradicts our goal** (i.e., better task generalization across multiple downstream tasks and datasets). If preference vectors are explored, some pretext tasks will definitely be jeopardized due to the definition of Pareto optimality.
> Such a bias over the pretext tasks would cause the self-supervised GNNs not performing equally well on every downstream task and dataset. It hurts the overall average performance and task generalization, which are the main focuses of this paper.
> >
> > **[Additional supporting experiments]** To empirically validate that an utilization of a preference vector contradicts our goal, we also conduct experiments on ParetoMTL [1] with five preference vectors (i.e., five vectors for different preferences over five tasks), as shown in the table below (Table 8 in the updated manuscript).
>
> | Dataset | w/o Pareto | **ParetoGNN** | ParetoMTL FeatRec | ParetoMTL TopoRec | ParetoMTL RepDecor | ParetoMTL MI-NG | ParetoMTL MI-NSG |
> |:-------:|:----------:|:---------:|:-----------------:|:-----------------:|:------------------:|:---------------:|:----------------:|
> | Wiki.CS |    74.64   | **76.03** |       74.15       |       72.42       |        70.36       |      72.44      |       75.89      |
> |  PubMed |    69.82   | **72.48** |       70.66       |       67.92       |        67.59       |      68.30      |       69.01      |
>
> [Table Re1: The average performance of ParetoGNN and ParetoMTL with preference vectors favoring different tasks.]
> > From Table Re1, we observe that multi-task self-supervised learning via Pareto learning with a preference vector under most of the cases does not outperform the vanilla weighted summation (i.e., w/o Pareto) due to the bias introduced by the preference vectors. Our proposed ParetoGNN consistently outperforms variants based on ParetoMTL with different preferences. Including preferences in multi-task learning could be very helpful when one or a set of training tasks need highlighting, whose success is demonstrated by the great experiments in ParetoMTL; but this is not our case as we emphasize the task generalization for GNNs. Our implementation of ParetoMTL comes from its official Github repo [3].

---

### Public Comment · ~Benedek_Andras_Rozemberczki1 · 2022-11-05
**Misattribution of datasets**

The paper misattributed the authorship of the Chameleons and Squirrels datasets. These datasets were proposed in this ICLR submission:

https://openreview.net/forum?id=HJxiMAVtPH

The Pei et al. paper cited by the authors took the Squirrel and Chameleons datasets and used those for benchmarking, but had nothing to do with the creation of the datasets. The correct citation for the paper which proposed the datasets is:

```bibtex
>@article{musae,
          author = {Rozemberczki, Benedek and Allen, Carl and Sarkar, Rik},
          title = {{Multi-Scale Attributed Node Embedding}},
          journal = {Journal of Complex Networks},
          volume = {9},
          number = {2},
          year = {2021},
}
```

---

> ### Author Response · Authors · 2022-11-12
> **Response to your concern**
>
> Dear Dr. Rozemberczki,
>
> Thanks for pointing out this issue. We have corrected the corresponding citation. We sincerely appreciate your contribution to the graph learning community by publicizing good datasets we utilized in this paper!
>
> Best regards,\
> ParetoGNN authors

---

### Author Response · Authors · 2022-11-12
**General Response to Reviewers**

We thank the reviewers for their feedback and constructive suggestions. We are pleased that all reviewers appreciated the contributions of our work, said "extensive experiments over four downstream tasks and eleven benchmark datasets" (WfQN), "a reasonable approach which is novel" (15ww), "well-written and organized" (wz1H), and unanimously agreed that our study of the task generalization over multiple downstream tasks and datasets in graph representation learning was insightful.

At the same time, multiple reviewers expressed some concerns about the novelty of our proposed Pareto learning. While it is true that our proposed task reconciliation by promoting Pareto optimality is "agnostic of domains" (WfQN), we want to emphasize that designing a multi-task self-supervised learning framework to enhance the task generalization of GNNs is not trivial, and that our proposed Pareto learning algorithm elegantly and effectively reconciles pretext tasks spanning different philosophies to further improve the task generalization with a convergence guarantee, which is not only technically but also philosophically novel.

While algorithmic and architectural innovations are critical for driving progress in the field, robust and insightful empirical investigations and studies of different aspects of performance (i.e., task generalization over different downstream tasks and datasets) are also essential for advancing our collective understanding of the existing research gaps.

An important contribution of our work is to analyze the task generalization of the existing SSL-based GNNs in a more rigorous setting. By conducting experiments over four downstream tasks and eleven benchmarks, we empirically demonstrated that the promising performance of existing SSL-based GNNs on one task or two usually does not translate into good task generalization. This observation could bring people's attention to the problems underlying the current trending evaluations of SSL-based GNNs and further shed light on potential solutions (like the multi-task self-supervised learning we explored in this work).
Aided with multi-task self-supervised learning that promotes Pareto optimality, ParetoGNN achieves the best task generalization by comparing with seven state-of-the-art competitive baselines. We believe that the comprehensive and robust experiments that we have provided for different SSL-based GNNs across various downstream tasks and datasets are extremely important resources to the community.

We also improved our manuscript by leveraging valuable comments from the reviewers. The modifications we made are listed as follows:
* With respect to the comments from Reviewer wz1H and WfQN, we added an extra section in the appendix (i.e., Appendix H in page 19) to further clarify the connections between ParetoGNN and Pareto learning frameworks in other domains.
* With respect to the comments from Reviewer 15ww, we clarified a question we asked in the introduction section, rephrased a conclusion we drew, and added reference/discussion with a missing related work.
* We also corrected a misattribution of datasets as suggested by the public comments.

We modified the manuscript to clarify your concerns, and we hope we have satisfactorily answered your questions. If so, could you please consider increasing your rating? If you have remaining doubts and concerns, please let us know, and we will happily respond. Thank you!

Best regards,\
ParetoGNN authors

---

### Author Response · Authors · 2022-11-28
**Reminder about Rebuttal**

Dear reviewers,

We thank you again for the thoughtful reviews and comments.

We hope we have satisfactorily answered your questions. If so, could you please consider increasing your rating? If you have remaining doubts and concerns, please let us know, and we will happily respond.

Thank you!

Best regards, \
ParetoGNN authors

---

### Decision · Program_Chairs · 2023-01-20

**Decision:**

Accept: poster

**Justification For Why Not Higher Score:**

All reviewers agree that the paper in question is above the threshold of acceptance. Unanimously, the paper reviewers graded the score of 6 which a slightly above the threshold of acceptance. This being said, the paper has the potential to be of great benefit to the community and its value can be appreciated by researchers investigating the problems addressed in the submission.

**Justification For Why Not Lower Score:**

All reviewers recommend the acceptance of the paper. The proposed method has the potential to be of great benefit to the community.

**Metareview: Summary, Strengths And Weaknesses:**

I. Summary:

- I.1 Investigated Problem:
The paper investigated the problem of task generalization for self-supervised learning (SSL) based graph neural networks (GNNs) where it shows that the promising performance on 1 or 2 tasks of SSL-based GNNs does not translate into good task generalization across various downstream tasks and datasets. ParetoGNN is then introduced for node representation over graphs.

- I.2 Proposed Solution:
The proposed method (ParetoGNN) is self-supervised by manifold pretext tasks that observe multiple philosophies. These multiple philosophies are reconciled using a multiple-gradient descent algorithm promoting Pareto optimality. The algorithm captures patterns that are relevant to several pretext tasks which reduce potential conflicts and in this way, knowledge is transferable to multiple tasks on graphs.

- I.3 Validity Proof of the Proposed Solution:
Both theoretical and empirical evidence is provided to support the validity of the proposed solution as a convergence guarantee of the presented algorithm is provided as well as extensive experiments over four downstream tasks and 11 benchmark datasets. The showcased evidence demonstrates that the presented solution can improve the model's task generalization ability.

II. Strengths:

- II.1 From a structural point of view: Reviewers agreed upon the organization and writing quality.
- II.2 From an analytical point of view: Reviewers pointed out the clarity of the submitted paper from a general perspective. More precisely. Reviewers appreciated:
    . The clarity of the motivation;
    . The presentation of the designed method;
    . The theoretical and empirical evidence provided to support the case of the proposed solution;
    . The discussion related to the comparison conducted with several features of the existing methods with the proposed method. In fact, it has been shown that the presented solution is comparable to or better than most baselines;
    . The transparency aspect of the submission as open source code is provided for reproducibility purposes and hyperparameters settings are detailed in the appendix.
- II.3 From a perspective of soundness (development, unity, and coherence) and completeness (correctness): The strength points mentioned above are sufficient evidence of the soundness and completeness of the paper. An additional point reinforcing the strength points mentioned above is the active interaction of the authors during the rebuttal period and their openness to concerns and questions raised by the reviewers. The openness followed by the active interactions and persistence in answering questions and addressing concerns related to the submission demonstrate the author's intellectual honesty and good faith in conveying the details of the proposed solution to the reader as well as its benefits. Such a rebuttal is a great example of how can rebuttals be conducted to reinforce a presented idea in a paper.

III. Addressing what can be thought of as weaknesses:

The weaknesses pointed out by the reviewers relate mostly to the potential aspect of the proposed method and its novelty.
- III.1 Before the rebuttal period, reviewers raised concerns about the novelty of the proposed method which is related to the fact that ParetoGNN cannot incorporate pre-defined trade-offs between pretext tasks. The authors addressed these concerns by explaining in which context it is useful to use preference vectors. The main focus of the paper is task generalization and the authors provided arguments and empirical results to prove that the use of preference vectors would hurt generalization across datasets and tasks.
- III.2 Missing related work suggested by the reviewers has been added by the authors.
- III.3 Another point has been raised by a reviewer which is about the generic aspect of the proposed method in question and its analysis beyond GNN models. This was discussed with one of the reviewers during the AC-reviewers meeting and we agreed that any good / idea can be applied to more a general context. This can be considered as a strong benefit of the proposed approach as it can be investigated in other contexts of applications and the use of models.

Most of the points that could be thought of as weaknesses have been addressed and it has been clarified that what can be thought of as a weakness is not. Unanimously, the reviewers agree on the acceptance of the submission.

IV. Potential of the paper:

- IV.1 From a Potential perspective (Potential of the paper to the community): The proposed solution has a great potential to be of benefit to the whole community, especially the ones interested in multi-objective learning and graph neural networks. The extension and investigation to other models can also be of great benefit to the representation learning community in general.

**Note From Pc:**

if the above contains the word "oral" or "spotlight" please see: "oral" presentation means -> notable-top-5% and "spotlight" means -> notable-top-25%. As stated in our emails, we are disassociating presentation type from AC recommendations

**Summary Of Ac-Reviewer Meeting:**

The meeting was conducted with one of the reviewers and as mentioned above.